# Learning to Configure Separators in Branch-and-Cut

**Sirui Li**[*]
MIT
siruil@mit.edu

**Wenbin Ouyang**[*]
MIT
oywenbin@mit.edu

**Max B. Paulus**
ETH Zürich
max.paulus@inf.ethz.ch

**Cathy Wu**
MIT
cathywu@mit.edu

## Abstract

Cutting planes are crucial in solving mixed integer linear programs (MILP) as they facilitate bound improvements on the optimal solution. Modern MILP solvers rely on a variety of separators to generate a diverse set of cutting planes by invoking the separators frequently during the solving process. This work identifies that MILP solvers can be drastically accelerated by appropriately selecting separators to activate. As the combinatorial separator selection space imposes challenges for machine learning, we *learn to separate* by proposing a novel data-driven strategy to restrict the selection space and a learning-guided algorithm on the restricted space. Our method predicts instance-aware separator configurations which can dynamically adapt during the solve, effectively accelerating the open source MILP solver SCIP by improving the relative solve time up to 72% and 37% on synthetic and real-world MILP benchmarks. Our work complements recent work on learning to select cutting planes and highlights the importance of separator management.

## 1 Introduction

Mixed Integer Linear Programs (MILP) have been widely used in logistics [15], management [12], and production planning [17]. Modern MILP solvers typically employ a Branch-and-Cut (B&C) framework that utilizes a Branch-and-Bound (B&B) tree search procedure to partition the search space. As illustrated in Fig. 1, cutting plane algorithms are applied within each node of the B&B tree, tightening the Linear Programming (LP) relaxation of the node and improving the lower bound.

This paper presents a machine learning approach to accelerate MILP solvers. Modern MILP solvers implement various cutting plane algorithms, also referred to as *separators*, to generate cutting planes that tighten the LP solutions. Different separators have varying performance and execution times depending on the specific MILP instance. Typical solvers use simple heuristics to select separators, which can limit the ability to exploit commonalities across problem instances. While there is a growing body of work considering the 'branch' and 'cut' aspects of B&C [18, 35, 51, 42], profiling the open-source academic MILP solver SCIP [8], we find generating cutting planes through separators is a major contributor to the total solve time, and deactivating unused separators leads to faster solves and fewer B&B tree nodes. That is, a well-configured separator setup allows the selected cutting planes to more effectively tighten the LP solution, leading to fewer nodes in the B&B tree.

To our knowledge, the problem of how to leverage machine learning for this critical task of separator configuration, namely the selection of separators to activate and deactivate during the MILP solving process has not been considered. Therefore, the goal of this paper is to explore the extent to which

---

[*]Equal Contribution

37th Conference on Neural Information Processing Systems (NeurIPS 2023).

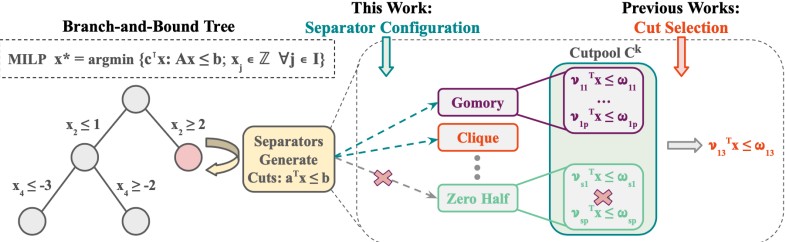

Figure 1: **Separator Configuration in Branch-and-Cut (B&C).** Modern MILP solvers perform Branch-and-Bound (B&B) tree search to solve MILPs. At each node of the B&B tree, cuts are added to tighten the Linear Programming (LP) relaxation of the MILP. To generate these cuts, a set of separators (e.g. Gomory) are invoked to first generate cuts into the cutpool $\mathcal{C}^k$. A subset of these cuts $\mathcal{P}^k \subseteq \mathcal{C}^k$ are then selected and added to the LP. The process is repeated for several separation rounds at each node. While previous works study cut selection from a pre-determined cutpool, this work focuses on the upstream task of separator configuration to generate a high quality cutpool efficiently.

tailoring the separator configuration to the MILP instance in a data-driven manner can accelerate MILP solvers. The central challenge comes from the high dimensionality of the configuration search space (induced by the large number of separators and configuration steps), which we address by introducing a data-driven search space restriction strategy that balances model fitting and generalization. We further propose a learning-guided algorithm, which is cast into the framework of neural contextual bandit, as an effective means of optimizing configurations within the reduced search space.

Our contributions can be summarized as

- We identify separator management as a crucial component in B&C, and introduce the Separator Configuration task for selecting separators to accelerate solving MILPs.
- To overcome the high dimensionality of the configuration task, we propose a data-driven strategy, directly informed by theoretical analysis, to restrict the search space. We further design a learning method to tailor instance-aware configurations within the restricted space.
- Extensive computational experiments demonstrate that our method achieves significant speedup over the competitive MILP solver SCIP on a variety of benchmark MILP datasets and objectives. Our method further accelerates the state-of-the-art MILP solver Gurobi and uncovers known facts from literature regarding separator efficacy for different MILP classes.

## 2 Related Work

The utilization of machine learning in MILP solvers has recently gained considerable attention. Various components in the B&B algorithm have been explored, including node selection [25, 49, 35], variable selection [18, 23, 58], branching rule [30, 23, 58, 46], scheduling primal heuristics [31, 26, 13], and deciding whether to apply Dantzig-Wolfe decomposition [34].

Our work is closely related to cutting plane selection, which can be achieved through heuristics [55, 2] or machine learning [51, 42]. The key difference, as shown in Figure 1 in Sec. 4, is that these works focus on selecting cutting planes from a pre-given cutpool generated by the available separators. That is, they consider the 'how to cut' question, whereas we focus on the equally crucial, but much less explored 'when (and what separators should we use) to cut' question [14, 16, 7]. For example, Wesselmann and Stuhl [55] state that they do not use any additional scheme to deactivate specific separators. In contrast, our work configures separators to generate a high-quality cutpool.

Another closely related line of work is on algorithm selection and parameter configurations [56, 57, 4, 5, 27, 28]. The most relevant works [57, 5] consider portfolio-based algorithm selection by first choosing a subset of algorithm parameter settings, and then selecting a parameter setting for each problem instance from the portfolio. We specialize and extend the general framework to separator configuration, by proposing a novel data-driven subspace restriction strategy, followed by a learning method, to configure separators for multiple separation rounds. We further present a theoretical analysis that directly informs our subspace restriction strategy, whereas the generalization guarantees from the prior work [5] is not informative for designing the portfolio-construction procedure.

It is common to restrict combinatorial space to improve the quality of solutions in discrete optimization. Previous research focuses primarily on decomposing large-scale problems, including heuristic works on Bender decomposition [44] and column generation [6], and recent learning-based works [50, 36] that train networks to select among a set of random or heuristic decomposition strategies. Our data-driven action space restriction strategy is general and could be of interest for a broader set of combinatorial optimization tasks, as well as other applications such as recommendation systems.

## 3 MILP Background

**Mixed Integer Linear Programming (MILP).** A MILP can be written as $x^* = \arg\min\{c^\mathsf{T} x : Ax \leq b, \ x_j \in \mathbb{Z} \ \forall j \in I\}$, where $x \in \mathbb{R}^n$ is the set of $n$ decision variables, $A \in \mathbb{R}^{m \times n}$ and $b \in \mathbb{R}^m$ formulate the set of $m$ constraints, and $c \in \mathbb{R}^n$ formulates the linear objective function. $I \subseteq \{1, ..., n\}$ defines the integer variables. $x^* \in \mathbb{R}^n$ denotes the optimal solution to the MILP with an optimal objective value $z^*$.

**Branch-and-Cut.** State-of-the-art MILP solvers perform branch-and-cut (B&C) to solve MILPs, where a branch-and-bound (B&B) procedure is used to recursively partition the search space into a tree. Within each node of the B&B tree, linear programming (LP) relaxations of the MILP are solved to obtain lower bounds. B&C further invoke Cutting plane algorithms to tighten the LP relaxation.

**Cutting Plane Separation.** When the optimal solution $x^*_{LP}$ to the LP relaxation is not a feasible solution to the original MILP, the cutting plane methods aim to find valid linear inequalities $\nu^\mathsf{T} x \leq \omega$ (cuts) that separate $x^*_{LP}$ from the convex hull of all feasible solutions of the MILP. Cutting plane separation happens in rounds, where each round $k$ consists of the following steps (1) solving the current LP relaxation, (2) calling different separators to generate a set of cuts and add them to the cutpool $\mathcal{C}_k$, (3) select a subset of cuts $\mathcal{P}_k \subseteq \mathcal{C}_k$ and update the LP with the selected cuts. Detailed background information on separators in the B&C framework can be found in Appendix A.1.

## 4 Problem Formulation

Different separators are designed to exploit different structures of the solution polytope defined by the MILP instance. The solution polytope also varies at different separation rounds, as changes to the constraints (e.g. after a branch) lead to different structures and thus different effective separators. Moreover, multiple separators can combine to exploit more sophisticated structures. The inherently combinatorial nature of the problem hence presents a challenge in assigning the appropriate separators to each MILP instances. This work aims to enhance the MILP solving process via intelligent separator configuration. We formally introduce the separator configuration task as follows.

**Definition 1 (Separator Configuration).** Suppose the MILP solver implements $M$ different separator algorithms. Given a set $\mathcal{X}$ of $N$ MILP instances (where $|\mathcal{X}| = N$), and a maximum number of separation rounds $R$ in a MILP solving process, we want to select a configuration $s_{x,n} \in \{0, 1\}^M$ for each instance $x \in \mathcal{X}$ and separation round $1 \leq n \leq R$, where the $w^{th}$ entry of $s_{x,n}$ equaling one means we activate the $w^{th}$ separator in separation round $n$, and equaling zero means we deactivate the $w^{th}$ separator in the corresponding round.

Figure 1 illustrates the separator configuration task and highlights the difference between our task and the downstream cutting plane selection task in previous works [51, 42].

We measure the success of an algorithm for the separator configuration task by the relative time improvement from SCIP's default configuration. Denote a proposed configuration policy as $\pi : \mathcal{X} \to \prod_{n=1}^{R} \{0, 1\}^M$, where for each MILP instance $x \in \mathcal{X}$, we have $\pi(x) = \{s_{x,1}, ..., s_{x,R}\}$ as the proposed configurations. Let $t_\pi(x)$ be the solve time of instance $x$ using the configuration sequence $\pi(x)$ and $t_0(x)$ be the solve time using the default SCIP configuration (both to optimality or a fixed gap). We evaluate the effectiveness of $\pi$ by the relative time improvement

$$\Delta(\pi) := \mathbb{E}_{x \in \mathcal{X}}[\delta(\pi(x), x)] \ \text{ where } \ \delta(\pi(x), x) := (t_0(x) - t_\pi(x)) / t_0(x) \tag{1}$$

The search space for the separator configuration task is enormous, with a size of $N \times 2^{M \times R}$. SCIP contains $M = 17$ separators, and a typical solve run yields $R \geq 30$, making the task highly challenging. In the next section, we discuss our data-driven approach to finding high quality configurations.

# 5 Learning to Separate

Two sources of high dimensionality in the search space come from (1) combinatorial number $|O| = 2^M$ of configurations, where each element of $O := \{0,1\}^M$ is a combination of separators (e.g., Gomory, Clique) to activate, and (2) a large number of configuration updates that results in the $|O|^R$ factor. We address the first challenge in Sec. 5.1 by restricting the number of configuration options, and the second challenge in Sec. 5.2 by reducing the frequency of configuration updates. The resulting restricted search space allows efficient learning in Sec. 5.3 to find high quality customized configurations for each MILP instance, which we term as instance-aware configurations.

## 5.1 Configuration space restriction

For simplicity, we first consider a single configuration update such that we apply the same configuration for all separation rounds, and our goal is to learn an instance-aware configuration predictor $\tilde{f} : \mathcal{X} \to O$. That is, we set $s_{x,1} = ... = s_{x,R} = \tilde{f}(x)$ for each $x \in \mathcal{X}$; Sec. 5.2 discusses extensions to multiple configuration updates. To address the challenge of learning the predictor in the high dimensional space $O$, we constrain the predictor $\tilde{f}_A$ to select from a subset $A \subseteq O$ of configurations with $|A|$ reasonably small, i.e. $\tilde{f}_A(x) \in A \; \forall x \in \mathcal{X}$. We design a data-driven strategy, supported by theoretical rationale, to identify a subspace $A$ for $\tilde{f}_A$ to achieve high performance.

**Preliminary definitions.** Let $\mathcal{X}$ be a class of MILP instances, and $\mathcal{K} = \{x_1, ..., x_K\} \subseteq \mathcal{X}$ be a given training set where we can acquire the time improvement $\{\delta(s, x_i)\}_{i=1..K; s \in O}$. The true performance of $\tilde{f}_A$ on $\mathcal{X}$ is $\Delta(\tilde{f}_A) = \mathbb{E}_{x \in \mathcal{X}}[\delta(\tilde{f}_A(x), x)]$, and the empirical counterpart on $\mathcal{K}$ is $\hat{\Delta}(\tilde{f}_A) = \frac{1}{K} \sum_{i=1}^{K} \delta(\tilde{f}_A(x_i), x_i)$. We further denote the true instance-agnostic performance of applying a single configuration $s \in \{0,1\}^M$ to all MILP instances as $\bar{\delta}(s) = \mathbb{E}_{x \in \mathcal{X}}[\delta(s, x)]$, and the empirical counterpart as $\hat{\bar{\delta}}(s) = \frac{1}{K} \sum_{i=1}^{K} \delta(s, x_i)$. Appendix A.2.1 details all relevant definitions.

**Restriction algorithm.** To find a subspace $A$ that optimizes the true performance $\Delta(\tilde{f}_A)$ for the predictor $\tilde{f}_A$, we employ the following training performance v.s. generalization decomposition:

$$\Delta(\tilde{f}_A) = \underbrace{\hat{\Delta}(\tilde{f}_A)}_{\text{training perf.}} - \underbrace{(\hat{\Delta}(\tilde{f}_A) - \Delta(\tilde{f}_A))}_{\text{generalization}} \tag{2}$$

The first term measures how well $\tilde{f}_A$ performs on the training set $\mathcal{K}$, while the second term reflects the generalization gap of $\tilde{f}_A$ to the entire distribution $\mathcal{X}$. Notably, a similar trade-off exists in standard supervised learning [47], where regularizations are used to balance fitting and generalization by implicitly restricting the hypothesis class. Relatedly, in this problem, we can balance the two terms by explicitly restricting the *output space* of the predictor. Intuitively, a larger subspace $A$ can improve training performance (more configuration options to leverage), but hurt generalization (more options that could perform poorly on unseen instances). This intuition is formalized next.

First, since the second term in Eq. (2) is unobserved, the following proposition imposes assumptions that allow us to restrict the configuration space. A detailed proof can be found in Appendix A.2.2.

**Proposition 1.** Assume the predictor $\tilde{f}_A$, when evaluated on the entire distribution $\mathcal{X}$, achieves perfect generalization (i.e., zero generalization gap) with probability $1 - \alpha$; with probability $\alpha$, the predictor makes mistake and outputs a configuration $s \in A$ uniformly at random. Then, the trainset performance v.s. generalization decomposition can be written as $\Delta(\tilde{f}_A) = (1 - \alpha)\hat{\Delta}(\tilde{f}_A) + \alpha \frac{1}{|A|} \sum_{s \in A} \bar{\delta}(s)$.

As $\bar{\delta}(s)$ is also unobservable, we further rely on its empirical counterpart $\hat{\bar{\delta}}_t(s)$ (see Appendix A.2.2 for a discussion of the reduction) and select the subspace $A$ based on the following objective:

$$(1 - \alpha)\hat{\Delta}(\tilde{f}_A) + \alpha \frac{1}{|A|} \sum_{s \in A} \hat{\bar{\delta}}(s) \tag{3}$$

The impact of the subspace $A$ on these two terms further depends on the nature of $\tilde{f}_A$; we assume that the predictor $\tilde{f}_A$ uses empirical risk minimization (ERM) and performs optimally on the training set $\mathcal{K}$, i.e. $\tilde{f}_A^{ERM}(x) = \arg\max_{s \in A} \delta(s, x) \; \forall x \in \mathcal{K}$, hence bypassing the need to train any predictor for constructing $A$. The discussion of the ERM assumption's validity and the extension to predictors with training error are provided in Appendix A.2.4 (See Lemma 3 for the extension).

Eq. (3) then sheds light on how to construct a good $A$ under the ERM assumption: an ideal subset $A$ allows $\tilde{f}_A^{ERM}$ to have (1) high training performance $\hat{\Delta}(\tilde{f}_A^{ERM})$, obtained when *some* configuration in $A$ achieves good performance for any MILP instance in a training set, and (2) low generalization gap, achieved when *each* configuration in $A$ has good performance across MILP instances in a test set, which we approximate with the average instance-agnostic performance on the training set $\frac{1}{|A|}\sum_{s\in A}\hat{\bar{\delta}}(s)$. In fact, a larger or more diverse subspace $A$ results in better $\hat{\Delta}(\tilde{f}_A^{ERM})$, as the ERM predictor can leverage more configuration options to improve the training set performance. Meanwhile, it may also lower $\frac{1}{|A|}\sum_{s\in A}\hat{\bar{\delta}}(s)$ which harms generalization, as we may include some configurations that perform poorly on most MILP instances but well on a small subset. The following proposition (proven in Appendix A.2.3) formalizes the diminishing marginal returns of $\tilde{f}_A^{ERM}$'s training performance with respect to $A$, which enables an efficient algorithm to construct $A$:

**Proposition 2.** The empirical performance of the ERM predictor $\hat{\Delta}(\tilde{f}_A^{ERM})$ is monotone submodular, and a greedy strategy where we include the configuration that achieves the greatest marginal improvement $\arg\max_{s\in\{0,1\}^M\setminus A}\hat{\Delta}(\tilde{f}_{A\cup\{s\}}^{ERM}) - \hat{\Delta}(\tilde{f}_A^{ERM})$ at each iteration is a $(1-1/e)$-approximation algorithm for constructing the subspace $A$ that optimizes $\hat{\Delta}(\tilde{f}_A^{ERM})$.

To balance the two terms in Eq. (3), we couple the greedy selection strategy with a filtering criterion that eliminates configurations with poor instance-agnostic performance to construct the subspace $A$. Due to the high computational cost of calculating the marginal improvement for all $2^M$ configurations, we first sample a large set $S$ of configurations, which we use to construct the subspace $A$. Then, at each iteration, we expand the current set $A$ with the configuration that produces the best marginal improvement in training performance, but only considering configurations $s \in S$ whose empirical instance-agnostic performance is greater than a threshold, i.e. $\hat{\bar{\delta}}(s) > b$. The extra filtering procedure enables us to improve the second term with small concessions in the first term. We continue the process while monitoring the two opposing terms, and terminate with a reasonably small $A$ that balances the trade-off. The detailed algorithm and discussions of the filtering and termination procedure are provided in Appendix A.3.

## 5.2 Configuration update restriction

Learning to update configurations at each separation round is challenging due to cascading errors from a large number of updates. Instead, we periodically update the configuration at a few intermediate rounds and hold it fixed between updates: we perform $k \ll R$ updates at rounds $\{n_j\}_{j=1}^k$ with $1 \le n_j \le R$, and set $s_{x,n_j} = ... = s_{x,n_{j+1}-1}$ for each $1 \le j \le k$ and $x \in \mathcal{X}$. Fig. 7 (Left) in Appendix A.4 shows an example of the configuration update restriction with $k = 2$ and $R = 6$, where we also discuss the trade-off of $k$ in approximation v.s. estimation. We empirically find a small number of updates can already yield a decent time improvement (we set $k = 2$ in Experiment Sec. 6).

We use a forward training algorithm [45] to learn the configuration policy $\tilde{\pi}^{(k)}: \mathcal{X} \to \prod_{i=1}^k \{0,1\}^M$. The algorithm decomposes the sequential task into $k$ single configuration update tasks $\tilde{\pi}^{(k)} = \{\tilde{f}_{\theta^T}^m\}_{m=1}^k$, where each $\tilde{f}_{\theta^T}^j: \mathcal{X} \to \{0,1\}^M$ is a separate network for the $j$-th configuration update. As illustrated in Fig.7 (Right) of Appendix A.4, at each iteration, we fix the weights of the trained networks for earlier updates $\{\tilde{f}_{\theta^T}^m\}_{m=1}^{j-1}$, and train the network $\tilde{f}_\theta^j$ for the $j^{th}$ update. The detailed algorithm is provided in Alg. 2 of Appendix A.4. We incorporate the configuration space restriction in Sec. 5.1 by constraining each network $\tilde{f}_\theta^j$ to select configurations from a subset $A \subseteq O$, such that $\tilde{f}_\theta^j(x) \in A$ for all $x \in \mathcal{X}$. This reduces the search space from $N \times 2^{M\times R}$ to $N \times |A| \times k$, significantly easing the learning process. Notably, we construct the subspace $A$ once at the initial update for computational efficiency benefits, as it yields comparable performances to constructing a new subspace for each update. Further details and discussions can be found in Appendix A.7.2.

## 5.3 Neural UCB algorithm

Given the restricted configuration space $A$, we frame each configuration update as a contextual bandit problem with $A$ arms (configurations). Conditional on the context (a MILP instance $x \in \mathcal{X}$), each arm $s \in A$ has a reward (time improvement $\delta(s,x)$). We employ the neural UCB algorithm [59] to efficiently train a network $\tilde{f}_{\theta^0}^j(x,s): \mathcal{X} \times A \to \mathbb{R}$ to estimate the reward, where the confidence

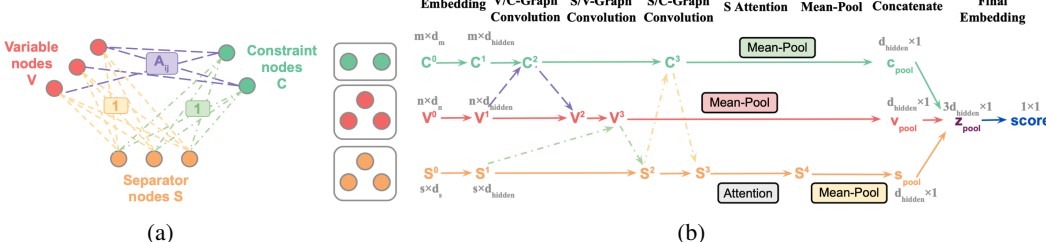

Figure 2: (a) **Our triplet graph encoding** of the MILP instance (the context) and the separator configuration (the arm / action). (b) **Our neural architecture** $\tilde{f}_\theta$. It involves three graph convolutions, an attention block for the separator nodes, and global poolings to extract the final score for reward prediction. We show the dimensionality of a tensor in gray if it is different from the previous size.

bound estimation is enabled by the small size of $A$. We provide the complete training procedure in Alg. 3 of Appendix A.5. At each training epoch $t$, we randomly sample $P$ instances from $\mathcal{X}$. For each instance, we sample $D$ configurations based on the upper confidence bound $ucb(x,s)$, which combines a reward point estimate $\tilde{f}^j_{\theta^{t-1}}(x,s)$ and a confidence bound estimate. The confidence bound estimate incorporates the gradient $\nabla_\theta \tilde{f}^j_{\theta^{t-1}}(x,s)$ and a normalizing matrix $Z_{t-1}$ only feasible to obtain when the number of arms is small. We run the MILP solver on each of the $P \times D$ pairs of instance-configuration and observe the reward labels. Lastly, we add all instance-configuration-reward tuples $\{x,s,r\}_{P \times D}$ to the data buffer and retrain the network $\tilde{f}^j_\theta$. At test time, we select the configuration with the highest predicted reward $s^*_{x,j} = \arg\max_{s \in A} \tilde{f}^j_{\theta^T}(x,s)$ or the highest UCB score $s^*_{x,j} = \arg\max_{s \in A} ucb(x,s)$, based on validation performance, at each update step $n_j$. We provide further details on the inference strategy in Appendix A.6.1.

**Context encoding.** We encode the context for each MILP instance $x$ and separator configuration $s$ as a triplet graph with three types of nodes in the graph: variable nodes $\mathbf{V}$, constraint nodes $\mathbf{C}$, and separator nodes $\mathbf{S}$. The variable and constraint nodes ($\mathbf{V}$, $\mathbf{C}$) appear in the previous works [18, 42]. We follow Paulus et al. [42] to use the same input features for $\mathbf{V}$ and $\mathbf{C}$, and construct edges between them such that a variable node $\mathbf{V}_i$ is connected to a constraint node $\mathbf{C}_j$ if the variable appears in the constraint with the weight corresponds to the coefficient $\mathbf{A}_{ij} \neq 0$. The separator nodes $\mathbf{S}$ are unique to our problem. We represent each configuration $s$ by $M$ separator nodes; each node $\mathbf{S}_k$ has $M+1$ dimensional input features, representing whether the separator is activated (the first dimension), and which separator it is (one-hot $M$-dimensional vector). We connect each separator node with all variable and constraint nodes, all with a weight of 1 for complete pairwise message passing. We provide detailed descriptions of the input features in Appendix A.5.

**Neural architecture** $\tilde{f}_\theta$. We extend the architecture in Paulus et al. [42] for our network $\tilde{f}_\theta(x,s)$ : $\mathcal{X} \times \mathcal{S} \to \mathbb{R}$. The architecture, as illustrated in Fig. 2, involves a Graph Convolutional Network (GCN) [33], an attention block on the hidden embeddings of the separator nodes [48], and a global pooling to output a single score for reward prediction. It first embeds $\mathbf{C}$, $\mathbf{V}$, and $\mathbf{S}$ input features into hidden representations, and performs message passing following the directions of $\mathbf{V} \to \mathbf{C} \to \mathbf{V}$, $\mathbf{S} \to \mathbf{V} \to \mathbf{S}$, and $\mathbf{S} \to \mathbf{C} \to \mathbf{S}$. Then, the $\mathbf{S}$ nodes pass through an attention module to emphasize the task of the separator configuration. Lastly, since we require the model to output a single score (in contrary to Paulus et al. [42] which outputs a score for each cut node), we perform a global mean pooling on each of the $\mathbf{C}$, $\mathbf{V}$, and $\mathbf{S}$ hidden embeddings to obtain three embedding vectors, concatenate them into a single vector, and finally use a multilayer perceptron (MLP) to map the vector into a scalar.

**Clipped Reward Label.** To account for variations in MILP solve time, we perform $l$ MILP solver runs for each configuration-instance pair $(s,x)$ and take the average time improvement as the unclipped reward label. Additionally, if a certain configuration $s$ takes significantly longer solve time than SCIP default on a MILP instance $x$, we terminate the MILP solver run when the relative time improvement is less than a predefined threshold $r_{\min} \ll 0$ to expedite data collection, and assign a clipped reward label of $r^{\mathrm{clip}}(s,x) = (\sum_{i=1..l} \max\{\delta^{(i)}(s,x), r_{\min}\})/l$. Reward clipping also simplifies learning by obviating the need to accurately fit the exact value of extreme negative improvements, which may skew the network's prediction. As long as the prediction's sign is right, we will not select such a configuration with a negative predicted value during testing.

Table 1: **Tang et al. and Ecole.** Absolute solve time of SCIP default, and the median (higher the better) and standard deviation (in parentheses) of relative time improvement of different methods.

| | | Tang | | | Ecole | | |
| --- | --- | --- | --- | --- | --- | --- | --- |
| | **Method** | Bin. Pack. | Max. Cut | Pack. | Comb. Auc. | Indep. Set | Fac. Loc. |
| | Default Time (s) | 0.076s (0.131s) | 1.77s (0.56s) | 8.82s (25.46s) | 2.73s (4.43s) | 8.21s (114.15s) | 61.1s (55.37s) |
| Heuristic Baselines | Default | 0% | 0% | 0% | 0% | 0% | 0% |
| | Random | -23.4% (153.8%) | -108.4% (168.2%) | -91% (127.8%) | -48.6% (159.0%) | -5% (161.4%) | -33.3% (157.2%) |
| | Prune | 13.9% (27.0%) | 2.7% (26.2%) | 6.6% (45.0%) | 12.3% (24.2%) | 18.0% (24.2%) | 24.7% (47.9%) |
| Ours Heuristic Variants | Inst. Agnostic Configuration | 33.7% (36.6%) | 69.8% (10.5%) | 20.1% (38.0%) | 60.1% (27.6%) | 57.8% (29.5%) | 11.5% (21.8%) |
| | Random within Restr. Subspace | 26.9% (33.6%) | 68.0% (11.0%) | 18.8% (38.7%) | 58.1% (28.7%) | 57.4% (75.8%) | 17.7% (33.0%) |
| Ours Learned | L2Sep | **42.3% (34.2%)** | **71.9% (11.3%)** | **28.5% (39.3%)** | **66.2% (26.2%)** | **72.4% (27.8%)** | **29.4% (39.6%)** |

**Loss function $\mathcal{L}$.** We use a $L_2$ loss between the prediction $\tilde{f}_\theta(x, s)$ and the clipped reward label $r^{\text{clip}}$:

$$L(\tilde{f}_\theta(x, s), r) = (\tilde{f}_\theta(x, s) - r^{\text{clip}})^2 \tag{4}$$

# 6 Experiments and Analysis

We divide the experiment section into two main parts. First, we evaluate our method on standard MILP benchmarks from Tang et al. [51] and Ecole [43], where the number of variables and constraints range from 60 to 10, 000. We conduct detailed ablation studies to validate the design choices made for our method. Second, we examine the efficacy of our method by applying it to large-scale real-world MILP benchmarks, including the MIPLIB [20], NN Verification [40], and Load Balancing in the ML4CO challenges [19], where the number of variables and constraints reaches up to 65, 000. We omit certain MILP classes from the benchmarks with excessively short solve times, few generated cutting planes, or small dataset sizes. Appendix A.6.5 provides a detailed description of the datasets.

## 6.1 Setup

**Evaluation Metric.** As we aim to accelerate SCIP solving through separator configuration, we evaluate our learned configuration by the relative time improvement from SCIP default, defined in Eq. (1), when both are solved to optimality (for standard instances) or a fixed gap (for large-scale instances) as described in Appendix A.6.4. We report the median and standard deviation across all test instances, and defer mean and interquartile mean to Appendix A.8 as they yield similar results.

**ML Setup.** We train the networks with ADAM [32] under a learning rate of $10^{-3}$. The reward label collection is performed via multi-processing with 48 CPU processes. As in previous works [51, 42, 54], we train separate models for each MILP class. By default, we generate a training set $\mathcal{K}_{small}$ of 100 instances for configuration space restriction, another training set $\mathcal{K}_{large}$ of 800 for predictor network training, a validation set of 100 instances, and a test set of 100 instances for each class Appendix A.6 provides full details of the setup.

**Baselines.** To our knowledge, our separator configuration task has not been explored in previous research. We design the following baselines to assess the effectiveness of our proposed methods: (1) **Default**, where we run SCIP with the default parameters; (2) **Random**, where for each MILP instance $x$, we randomly sample a configuration $s \in \{0, 1\}^M$; (3) **Prune**, where we first run SCIP default on the $\mathcal{K}_{small}$, and then at test time, we deactivate separators whose generated cutting planes are never applied to any instances in $\mathcal{K}_{small}$.

**Proposed Methods.** We evaluate the performances of our complete method and its sub-components: (1) **Ours (L2Sep)**, where we perform $k = 2$ instance-aware configuration updates per MILP instance

Table 2: **Detailed ablations of different components in our L2Sep algorithm.** Learning with neural UCB in the restricted config. space and performing $k = 2$ config. updates achieves the best result.

| Ablation Method | Config. Space Restriction | | Config. Update Restriction | | Neural Contextual Bandit | | Ours: L2Sep |
|---|---|---|---|---|---|---|---|
| | No Restr. | Greedy Restr. | $k = 1$ | $k = 3$ | Supervise ($\times 4$) | $\epsilon$-greedy | w/ Restr. + $k = 2$ + UCB |
| **Bin. Pack.** | 18.6% (125.3%) | 35.8% (35.6%) | 40.4% (42.3%) | 44.2% (32.4%) | 40.2% (19.7%) | 36.3% (32.3%) | 42.3% (34.2%) |
| **Pack.** | 19.6% (61.1%) | 18.4% (49.8%) | 23.8% (38.1%) | 27.8% (38.1%) | 24.0% (44.1%) | 25.1% (44.3%) | 28.5% (39.3%) |
| **Indep. Set** | 38.6% (23.5%) | 68.5% (28.1%) | 70.2% (38.6%) | 69.7% (29.1%) | 68.7% (33.9%) | 64.1% (48.7%) | 72.4% (27.8%) |
| **Fac. Loc.** | 15.5% (121.2%) | 27.1% (38.7%) | 20.1% (37.8%) | 29.7% (29.8%) | 31.0% (41.5%) | 28.1% (23.6%) | 29.4% (39.6%) |

(Sec. 5.2). We use forward training to learn predictors via the neural UCB algorithm (Sec. 5.3) within the restricted configuration subspace $A$ (Sec. 5.1). (2) **Instance Agnostic Configuration**, where we select a single configuration $\tilde{s}$ with the best instance-agnostic performance $\hat{\bar{\delta}}(\tilde{s})$ on $\mathcal{K}_{small}$ from the initial large subset $S$ for our space restriction algorithm ($|S| \approx 2000$); $\tilde{s}$ is included in $A$. (3) **Random within Restricted Subspace**, where for each MILP instance, we select a random configuration within $A$. The latter two sub-components assess the quality of the restricted subspace and the benefit of learning instance-aware configurations. Further details can be found in Appendix A.6.1.

## 6.2 Standard MILP Benchmarks with Detailed Ablations

**Performance.** Table 1 presents the relative time improvement of different methods over SCIP default, on the datasets of Tang et al. and Ecole. Our method demonstrates a substantial speed up from SCIP default across all MILP classes, with a relative time improvement ranging from 25% to 70%. In contrast, the random baseline performs poorly, demonstrating that separator configuration is a nontrivial task. Meanwhile, although the pruning baseline generally outperforms SCIP default, its time improvement is significantly less than ours, confirming the efficacy of our proposed algorithm. Notably, both of our two heuristic sub-components achieve impressive speed-up from SCIP default, indicating the high quality of our restricted subspace (and a configuration within) to accelerate SCIP; additionally, our complete learning method outperforms the sub-components on all MILP classes, further underscoring the advantages of learning for instance-aware configurations.

We note that the high standard deviation, exhibited in all methods including SCIP default and also observed in the recent studies [54], is reasonable due to instance heterogeneity, as the standard deviation is calculated based on the time improvements across instances within each MILP dataset.

**Ablations.** In Table 2, we further conduct comprehensive ablation studies to assess the effectiveness of our learning method. The ablations are performed on four representative MILP classes in Ecole and Tang, covering a wide range of problem sizes and solve times. Appendix A.7 provides detailed descriptions as well as additional ablation results. We aim to answer the following questions: (i) Does the restricted config. space improve learning performance? (ii) How does the performance vary with fewer or more updates? (iii) Does the use of neural UCB lead to efficient predictor learning?

**(i) Configuration space restriction (Sec. 5.1).** We train our configuration predictors to select within a restricted subspace $A$ constructed by a greedy strategy coupled with a filtering criterion. To evaluate the importance of the space restriction in learning high quality predictors, we perform an ablation study where we train the predictors to select within (1) the unrestricted space $O = \{0, 1\}^M$ (**No Restr.**), and (2) a same-sized subspace $A'$ constructed solely by the greedy strategy without filtering (**Greedy Restr.**). The restricted search space substantially enhances the learned predictors when compared to **No Restr.**, improving the median performance and lowering the standard deviation. We also observe the benefit of the filtering criterion when compared to **Greedy Restr.**. The filtering criterion excludes configurations with subpar instance-agnostic performance from entering the restricted configuration space, improving model generalization as demonstrated in our theoretical analysis.

Table 3: **Real-world MILPs.** Absolute solve time of SCIP default, and the median (higher the better) and standard deviation (in parentheses) of relative time improvement of different methods.

| Methods | Default Times (s) | Heuristic Baselinses | | | Ours Heuristic Variants | | Ours Learned |
| | | Default | Random | Prune | Inst. Agnostic Configuration | Random within Restr. Subspace | L2Sep |
| --- | --- | --- | --- | --- | --- | --- | --- |
| **MIPLIB** | 25.08s (57.05s) | 0% | -149.1% (149.7%) | 4.8% (107.6%) | 5.5% (71.5%) | 1.9% (74.9%) | **12.9%** **(73.1%)** |
| **NN Verification** | 31.42s (22.44s) | 0% | -300.0% (152.3%) | 31.5% (36.3%) | 31.4% (38.3%) | 30.7% (34.1%) | **37.5%** **(33.9%)** |
| **Load Balancing** | 31.86s (7.07s) | 0% | -300.1% (129.5%) | 21.1% (150.8%) | 10.4% (8.5%) | 10.0% (31.5%) | **21.2%** **(20.3%)** |

**(ii) Configuration update restriction (Sec. 5.2).** We apply the forward training algorithm (Sec. 5.2) to perform two configuration updates ($k = 2$) for each MILP instance. To examine the impact of fewer or more updates, we conduct an ablation study where we (1) performed a single update at round $n_1 = 0$ (**k = 1**), and (2) added an additional third update at a later round $n_3$ (**k = 3**). The results show that while a single update yields decent time improvement, adding the second update leads to further time savings. Meanwhile, we observe little improvement from the third update (**k = 3**). We speculate that this is because the performance improvement primarily occurs during the early stages of a solve, and holding a fixed configuration for longer may be advantageous by making the solve process more stable. We leave further investigation of more configuration updates as a future work.

**(iii) Neural UCB algorithm (Sec. 5.3).** Our method employs the online neural UCB algorithm to improve training efficiency for configuration predictors. We present the ablation (1) where we train the predictor using an offline regression dataset whose size is four times as ours while training the model until convergence (**Supervise ($\times 4$)**); we conduct an additional ablation (2) where we train the predictor using neural contextual bandit with $\epsilon$-greedy exploration strategy ($\epsilon$**-greedy**). Our model performs comparably to **Supervise ($\times 4$)** while using significantly fewer data, highlighting the importance of the contextual bandit for improving training efficiency by collecting increasingly higher quality datasets online. The ablation results with $\epsilon$**-greedy** further confirm the benefit of the confidence bound estimation in neural UCB for more efficient contextual bandit exploration.

## 6.3 Large-scale Real-world MILP Benchmarks

The real-world datasets of MIPLIB, NN Verification, and Load Balancing present significant challenges due to the vast number of variables and constraints (on the order of $10^4$), including nonstandard constraint types that MILP separators are not designed to handle. MIPLIB imposes a further challenge of dataset heterogeneity, as it contains a diverse set of instances from various application domains. Prior research [52] struggles to learn effectively on MIPLIB due to this heterogeneity, and a recent study [54] attempts to learn cutting plane selection over two homogeneous subsets (with 20 and 40 instances each). In contrast, we attempt to learn separator configuration across a larger MIPLIB subset that includes 443 of the 1065 instances in the original set, while carefully preserving the heterogeneity of the dataset. We provide our subset curation procedure in Appendix A.6.5.

**Main Results.** Table 3 presents the relative time improvement of various methods over SCIP default, on the large-scale real-world datasets. Again, our complete method displays a substantial speed up from SCIP default with a relative time improvement ranging from 12% to 37%. Our method also improves from our heuristic sub-components, further indicating the efficacy of our learning component on the challenging datasets. In contrast, the random baseline fails to improve from SCIP default, while the pruning baseline, despite having a reasonable median performance, suffers from a high standard deviation due to poor performance on many instances (See Appendix A.8.1 for IQM and mean results). Our results show the effectiveness of our learning method in improving the efficiency of practical applications that involve large-scale MILP optimization.

Although not a perfect comparison, for reference, we attempt to contextualize our result by examining the time improvement in the most comparable setting we found, which we provide comparison details in Appendix A.8.2: the learning method for cutting plane selection in Paulus et al. [42] achieves a median relative time improvement of $11.67\%$ on the NN Verification dataset, and that in Wang et al. [54] obtains a 3% and 1% improvement in the solve time on two small homogeneous MIPLIB

Table 4: **Gurobi as the MILP Solver.** Absolute solve time of Gurobi default, and the median (higher the better) and standard deviation (in parentheses) of relative time improvement of different methods.

| Methods | Default Times (s) | Heuristic Baselinses | | Ours Heuristic Variants | | Ours Learned |
| | | Default | Random | Inst. Agnostic Configuration | Random within Restr. Subspace | L2Sep |
|---|---|---|---|---|---|---|
| **Max. Cut** | 0.087s (0.051s) | 0% | 18.6% (49.0%) | 35.1% (35.8%) | 37.3% (48.0%) | **45.4% (38.4%)** |
| **Pack.** | 4.048s (3.216s) | 0% | 15.5% (28.2%) | 22.9% (39.4%) | 24.3% (32.2%) | **30.6% (29.6%)** |
| **Comb. Auc.** | 1.687s (3.596s) | 0% | -10.7% (69.1%) | 3.1% (65.3%) | 5.1% (84.2%) | **12.6% (63.5%)** |
| **Fac. Loc.** | 27.872s (14.733s) | 0% | 13.4% (46.0%) | 40.6% (48.1%) | 40.2% (46.8%) | **56.7% (35.7%)** |

subsets. While the comparison is far from perfect, our learning method for separator configuration achieves much higher time improvements of 37.5% on NN Verification and 12.9% on MIPLIB.

### 6.4 Interpretation Analysis: L2Sep Recovers Effective Separators from Literature

**Bin Packing:** It is known that instances with few bins approximate the Knapsack problem (Clique cuts are known to be effective [9]), and that instances with many bins approximate Bipartite Matching (Flowcover cuts can be useful [53]). We analyze the separators activated by L2Sep when we gradually decrease the number of bins, and observe that the prevalence of selected Clique and Flowcover cuts increased and decreased, respectively. This is illustrated in Fig 8 in Appendix A.8.3.

**Other MILP Classes:** We provide visualizations and interpretations for other MILP classes in Appendix A.8.3. Notably, Clique is known to be effective for Indep. Set [16]; L2Sep recovers this fact by frequently selecting configurations that activate Clique. Meanwhile, L2Sep discovers the instance heterogeneity of MIPLIB, resulting in a more dispersed distribution of selected configurations.

### 6.5 State-of-the-art MILP Solver Gurobi

We apply our method L2Sep with Gurobi, which contains a larger set of 21 separators. As Gurobi is closed-source, we cannot change configurations after the solving process starts, so we only consider one stage of separator configuration ($k = 1$). As seen from Table 4, L2Sep achieves significant relative time improvements over the Gurobi default, with gains ranging from 12% to 56%. This result confirms the efficacy of L2Sep as an automatic instance-aware separator configuration method.

### 6.6 Additional Results

In Appendix A.8.5 and A.8.4, we further demonstrate (1) Separator Configuration has immediate and multi-step effects in the B&C Process. For instance, even though L2Sep does not modify branching, the branching solve time is reduced. (2) L2Sep is effective under an alternative objective, achieving 15%-68% relative gap improvements under fixed time limits.

## 7 Conclusion

This work identifies the opportunity of managing separators to improve MILP solvers, and further formulates and designs a learning-based method for doing so. We design a data-driven strategy, supported by theoretical analysis, to restrict the combinatorial space of separator configurations, and overall find that our learning method is able to improve the relative solve time (over the default solver) from 12% to 72% across a range of MILP benchmarks. In future work, we plan to apply our algorithm to more challenging MILP problems, particularly those that cannot be solved to optimality. We also aim to learn more fine-grained controls by increasing the frequency of separation configuration updates. Our algorithm is highly versatile, and we plan to investigate its potential to manage aspects of the MILP solvers, and further integrate with previous works on cutting plane selection. Our code is publicly available at `https://github.com/mit-wu-lab/learning-to-configure-separators`. We believe that our learning framework can be a powerful technique to enhance MILP solvers.

## Acknowledgments and Disclosure of Funding

The authors would like to thank Mark Velednitsky and Alexandre Jacquillat for insightful discussions regarding an interpretative analysis of the learned model. This work was supported by a gift from Mathworks, the National Science Foundation (NSF) CAREER award (#2239566), the MIT Amazon Science Hub, and MIT's Research Support Committee. The authors acknowledge the MIT SuperCloud and Lincoln Laboratory Supercomputing Center for providing HPC resources that have contributed to the research results reported within this paper.

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
