# A   Supplementary Material: *Learning to Separate in Branch-and-Cut*

# Contents

## A.1 MILP and Branch-and-Cut Background

**Mixed Integer Linear Programming (MILP).** A MILP can be written as

$$x^* = \arg\min\{c^\intercal x : Ax \leq b, \ x_j \in \mathbb{Z} \ \forall j \in I\} \tag{5}$$

where $x \in \mathbb{R}^n$ is a set of $n$ decision variables, $A \in \mathbb{R}^{m \times n}$ and $b \in \mathbb{R}^m$ formulate a set of $m$ constraints, and $c \in \mathbb{R}^n$ formulates the linear objective function. $I \subseteq \{1, ..., n\}$ defines the variables that are required to be integral. $x^* \in \mathbb{R}^n$ denotes the optimal solution to the MILP with an optimal objective value $z^*$.

**Branch-and-Cut.** State-of-the-art MILP solvers perform branch-and-cut (B&C) to solve MILPs, where a branch-and-bound (B&B) procedure is used to recursively partition the search space into a tree. Within each node of the B&B tree, linear programming (LP) relaxations of Eq. 5 are solved to obtain lower bounds to the MILP. Specifically, a LP relaxation of Eq. (5) can be written as

$$x_{LP}^* = \arg\min\{c^\intercal x : Ax \leq b, \ x \in \mathbb{R}^n\} \tag{6}$$

where $x_{LP}^* \in \mathbb{R}^n$ denotes the optimal solution to the LP with an optimal objective value $z_{LP}^*$ such that $z_{LP}^* \leq z^*$.

**Cutting Plane Separation.** Each node of the B&B tree uses cutting plane algorithms to tighten the LP relaxation. When $x_{LP}^*$ in Eq. (6) does not satisfy $(x_{LP}^*)_j \in \mathbb{Z} \ \forall j \in I$, it is not a feasible solution to the original MILP. The cutting plane methods aim to find valid linear inequalities $\nu^\intercal x \leq \omega$ (cuts) that separate $x_{LP}^*$ from the convex hull of all feasible solutions of the MILP. Namely, a cut satisfies $\nu^\intercal x_{LP}^* > \omega$, and $\nu^\intercal x \leq \omega$ for each feasible solution $x$ to the MILP. Adding cuts into the LP tightens the relaxation, leading to a better lower bound to the MILP.

Cutting plane separation happens in rounds, where each separation round $k$ consists of the following steps (1) solving the current LP relaxation, (2) if separation conditions are satisfied, calling different separators to generate a set of cuts and add them to the cutpool $\mathcal{C}_k$, (3) select a subset of cuts $\mathcal{P}_k \subseteq \mathcal{C}_k$ and update the LP with the selected cuts.

Typical MILP solvers, such as SCIP [8] and Gurobi [24], maintain a set of separators such as Gomory [3] and Flow cover [22] to generate cuts. Each separator in SCIP has a priority and a frequency attribute, and once invoked, generates a set of cuts that are added to the cutpool $\mathcal{C}_k$. The frequency decides the depth level of the B&B tree node in which the separator is invoked (typically the root node and all other nodes with depth divisible by some constant $d$). The priority decides the order of the separators to be invoked; in each separation round, separators are invoked with a descending priority order until a predefined maximal number of cuts $max_{cuts}$ are generated. Separators with low priority may not be invoked during a separation round. By default, the priorities and frequency attributes in SCIP are a set of predefined values that remain unchanged for all MILP instances.

**Benefits of the CutPool $\mathcal{C}_k$.** MILP solvers do not directly add all cuts generated by the separators to the MILP, as adding a large number of cuts increases the MILP size and slows down the solver. Instead, a cutpool $\mathcal{C}_k$ is used as an intermediate buffer to hold a diverse set of cuts generated by a variety of separators. The cutting plane selector can then compare the cuts in the cutpool and select the most effective ones for the current stage of the MILP solve. Thus, well-designed separator configurations can not only expedite cutting plane generation by deactivating time consuming separators, but can also yield a superior quality cutpool $\mathcal{C}_k$ that may in turn enhance the performance of cutting plane selection $\mathcal{P}_k$ that leads to a further reduction the MILP solve time.

Cutpool has an additional advantage of storing previously generated cuts for future separation rounds and branch-and-cut tree nodes, thereby saving time by reducing the number of calls to expensive separators [1]. As a consequence, configuring separators at a separation round can have both an immediate and long-term impact on the branch-and-bound process due to the presence of the cutpool.

## A.2 Configuration Space Restriction: Proofs and Discussions

### A.2.1 Preliminary definitions

Table 5: Definition table for key terms used in the paper, including the true and empirical performance of instance-aware predictor and instance-agnostic configuration, and the optimal and empirical risk minimizing (ERM) predictor and configuration on both the original space and the restricted subspace. The relative time improvement $\delta$ is defined in Eq. (1) of the main paper. We consider a single configuration update per MILP instance, as in Sec 5.1 of the main paper. The unrestricted space is denoted as $O = \{0,1\}^M$.

| | | Instance-aware predictor | Instance-agnostic configuration |
|---|---|---|---|
| **True** $\mathcal{X}$ | Perf. Measure | $\Delta(f) = \underset{x \in \mathcal{X}}{\mathbb{E}} [\delta(f(x), x)]$ | $\bar{\delta}(s) = \underset{x \in \mathcal{X}}{\mathbb{E}} [\delta(s, x)]$ |
| | Optimal Action (unobserved) | function $f_O^* \to \{0,1\}^M$ s.t. $f_O^*(x) = \underset{s \in \{0,1\}^M}{\arg\max} \delta(s,x) \ \forall x \in \mathcal{X}$ | config. $s_O^* \in \{0,1\}^M$ s.t. $s_O^* = \underset{s \in \{0,1\}^M}{\arg\max} \bar{\delta}(s)$ |
| | Optimal *Subspace-restricted* Action (unobserved) | function $f_A^* \to A \subseteq \{0,1\}^M$ s.t. $f_A^*(x) = \underset{s \in A}{\arg\max} \delta(s,x) \ \forall x \in \mathcal{X}$ | config. $s_A^* \in A \subseteq \{0,1\}^M$ s.t. $s_A^* = \underset{s \in A}{\arg\max} \bar{\delta}(s)$ |
| **Empirical** $\mathcal{K}$ | Perf. Measure | $\hat{\Delta}(f) = \frac{1}{K} \sum_{i=1}^{K} \delta(f(x_i), x_i)$ | $\hat{\bar{\delta}}(s) = \frac{1}{K} \sum_{i=1}^{K} \delta(s, x_i)$ |
| | ERM Action (observed) | predictor $\tilde{f}_O^{ERM} : \mathcal{X} \to \{0,1\}^M$ s.t. $\tilde{f}_O^{ERM}(x) = \underset{s \in \{0,1\}^M}{\arg\max} \delta(s,x) \ \forall x \in \mathcal{K}.$ | config. $\tilde{s}_O^{ERM} \in \{0,1\}^M$ s.t. $\tilde{s}_O^{ERM} = \underset{s \in \{0,1\}^M}{\arg\max} \hat{\bar{\delta}}(s)$ |
| | ERM *Subspace-restricted* Action (observed) | predictor $\tilde{f}_A^{ERM} : \mathcal{X} \to A \subseteq \{0,1\}^M$ s.t. $\tilde{f}_A^{ERM}(x) = \underset{s \in A}{\arg\max} \delta(s,x) \ \forall x \in \mathcal{K}.$ | config. $\tilde{s}_A^{ERM} \in A \subseteq \{0,1\}^M$ s.t. $\tilde{s}_A^{ERM} = \underset{s \in A}{\arg\max} \hat{\bar{\delta}}(s)$ |

**True performance.** Let $\mathcal{X}$ be a class of MILP instances. Let $f : \mathcal{X} \to \{0,1\}^M$ be a configuration function. The true performance of $f$ is defined as $\Delta(f) = \underset{x \in \mathcal{X}}{\mathbb{E}} [\delta(f(x), x)]$.

The optimal configuration function $f_O^* \to \{0,1\}^M$ is $f_O^*(x) = \underset{s \in O}{\arg\max} \delta(s,x) \ \forall x \in \mathcal{X}$, and the optimal subspace-restricted configuration function $f_A^* \to A \subseteq \{0,1\}^M$ is $f_A^*(x) = \underset{s \in A}{\arg\max} \delta(s,x) \ \forall x \in \mathcal{X}$.

During training, we do not have access to the optimal configuration function nor the time improvement for all configurations and MILP instances in $\mathcal{X}$. Instead, we are given a set of training instances $\mathcal{K}$, from which we can collect the time improvements for different configurations by calling the MILP solver, to learn a configuration predictor. We define the predictor's empirical performance on the training instances as follows.

**Empirical performance.** Let $f : \mathcal{X} \to \{0,1\}^M$ be a configuration predictor. Let $\mathcal{K} = \{x_1, ..., x_K\}$ be a set of training MILP instances. The empirical performance of $f$ on $\mathcal{K}$ is $\hat{\Delta}(f) = \frac{1}{K} \sum_{i=1}^{K} \delta(f(x_i), x_i)$.

Given a subspace $A \subseteq \{0,1\}^M$, an empirical risk minimization (ERM) configuration predictor $\tilde{f}_A^{ERM} : \mathcal{X} \to A$ selects the best configuration within $A$ for each instance in the training set $\mathcal{K}$, i.e. $\tilde{f}_A^{ERM}(x) = \underset{s \in A}{\arg\max} \delta(s,x) \ \forall x \in \mathcal{K}$. That is, $\tilde{f}_A^{ERM}(x)$ coincides with $f_A^*$ on $\mathcal{K}$.

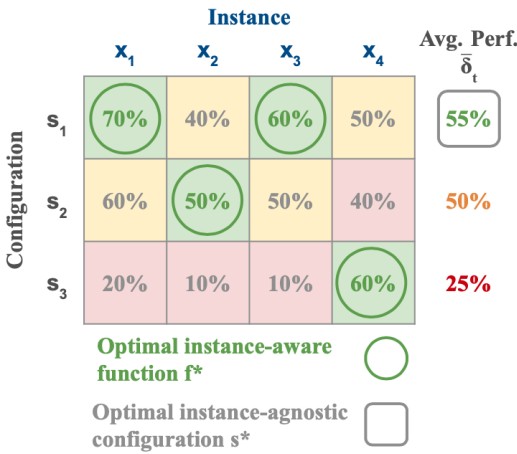

Figure 3: An illustration of the difference between the **optimal instance-aware function** $f^*$, which maps each MILP instance to a (possibly different) configuration that maximizes the time improvement, and the **optimal instance-agnostic configuration** $s^*$, which is a single configuration that achieves the highest average time improvement across the instances.

**Instance-agnostic performance.** For each configuration $s \in \{0, 1\}^M$, We further denote the true instance-agnostic performance of applying the same $s$ to all MILP instances as $\bar{\delta}(s) = \underset{x \in \mathcal{X}}{\mathbb{E}} [\delta(s, x)]$,

and the corresponding empirical instance-agnostic performance as $\hat{\bar{\delta}}_t(s) = \frac{1}{K} \sum_{i=1}^{K} \delta(s, x_i)$.

The optimal and ERM instance-agnostic configuration is defined as $s_O^* = \underset{s \in O}{\arg\max}\, \bar{\delta}_t(s)$ and $\tilde{s}_O^{ERM} = \underset{s \in O}{\arg\max}\, \hat{\bar{\delta}}_t(s)$, and the optimal and ERM instance-agnostic configuration on a restricted subspace $A \subseteq \{0, 1\}^M$ is similarly defined as $s_A^* = \underset{s \in A}{\arg\max}\, \bar{\delta}_t(s)$ and $\tilde{s}_A^{ERM} = \underset{s \in A}{\arg\max}\, \hat{\bar{\delta}}_t(s)$. Table 5 provides a list of all related concepts, and Fig. 3 illustrates the difference between the optimal instance-aware function and the optimal instance-agnostic configuration.

### A.2.2  Proof of Proposition 1

**Proposition 1.**  Assume that a configuration predictor $\tilde{f}_A$, when evaluated on the entire distribution $\mathcal{X}$, achieves perfect generalization (i.e., zero generalization gap) with probability $1 - \alpha$. With probability $\alpha$, the predictor makes mistakes and outputs a configuration $s \in A$ uniformly at random. Then, the trainset performance v.s. generalization decomposition can be written as

$$\Delta(\tilde{f}_A) = (1 - \alpha)\hat{\Delta}(\tilde{f}_A) + \alpha \frac{1}{|A|} \sum_{s \in A} \bar{\delta}(s) \tag{7}$$

**Proof.** By definition, we have $\Delta(\tilde{f}_A) = \mathbb{E}_{x \in \mathcal{X}} [\delta(f(x), x)]$. From the assumption, we have

$$\Delta(\tilde{f}_A) = \begin{cases} \hat{\Delta}(\tilde{f}_A) & \text{with probability } \alpha \\ \frac{1}{A} \sum_{s \in A} \mathbb{E}_{x \in \mathcal{X}} [\delta(s, x)] = \frac{1}{A} \sum_{s \in A} \bar{\delta}(s) & \text{with probability } 1 - \alpha \end{cases} \tag{8}$$

Hence, from Eq. (2) of the main paper, we get

$$\begin{aligned} \Delta(\tilde{f}_A) &= \hat{\Delta}(\tilde{f}_A) - \left( \hat{\Delta}(\tilde{f}_A) - \Delta(\tilde{f}_A) \right) \\ &= \hat{\Delta}(\tilde{f}_A) - (1 - \alpha) \cdot 0 - \alpha \cdot \left( \hat{\Delta}(\tilde{f}_A) - \frac{1}{A} \sum_{s \in A} \bar{\delta}(s) \right) \\ &= (1 - \alpha)\hat{\Delta}(\tilde{f}_A) + \alpha \frac{1}{A} \sum_{s \in A} \bar{\delta}(s) \quad \blacksquare \end{aligned} \tag{9}$$

**Assumption Discussion (Generalization error).**  The second average instance-agnostic performance term is a result of the assumption that the predictor selects a configuration randomly when it makes a mistake. In practice, the predictor's performance could be worse. For example, the predictor may select the configuration with the poorest instance-agnostic performance. In such a scenario, our algorithm's filtering strategy (See Alg. 1) that excludes configurations with an average performance below a threshold $\hat{\bar{\delta}}(s) \leq b$ remains highly beneficial: with this strategy, we can ensure that the performance of all selected configurations, including the worst one, is above the threshold value of $b$. Moreover, when deciding the size of the subspace $A$, we can track the performance of the worst selected configuration in addition to the average performance across all selected configurations to account for situations where the predictor's mistakes lead to worst-case performance.

**Assumption Discussions (Empirical instance-agnostic perf.).**  In Eq. (3) of the main paper, we approximate the true instance-agnostic performance of each configuration $\bar{\delta}(s)$ by the empirical counterpart $\hat{\bar{\delta}}(s)$, under the assumption that different configuration $s$ have similar generalization behavior when we apply each configuration to all instances. We test the generalization of different configurations by sampling a hold-out validation set $\mathcal{V}$, and compare the performance of $\hat{\bar{\delta}}(s)$ evaluated on the training set $\mathcal{K}_{small}$ (denoted as $\hat{\bar{\delta}}^{\mathcal{K}_{small}}(s)$) and on the hold-out set $\mathcal{V}$ (denoted as $\hat{\bar{\delta}}^{\mathcal{V}}(s)$).

The scatter plots in Fig. 4 show the instance-agnostic performances for Maximum Cut and Independent Set on a training set $\mathcal{K}_{small}$ of 100 instances and a hold-out set $\mathcal{V}$ of 100 instances. The plotted configurations are selected from the initial configuration space $S$ (see Alg. 1) by picking from each bin in the histogram of the set $\{\hat{\bar{\delta}}^{\mathcal{K}_{small}}(s), s \in S\}$ to ensure a diverse range of instance-agnostic performances among the chosen configurations. The darkness and size of each circle (configuration) in the plot are proportional to the total number of configurations in the corresponding bin, divided by the number of samples selected from that bin.

The strong linear trend $y = x$ observed in each scatter plot, along with the perfect alignment of configurations excluded by the filtering strategy in Alg. 1 (represented by circles in the bottom left corner split by the black dotted lines) validate our approximation of the true instance-agnostic performance with the empirical counterpart.

Notably, while we observe a strong linear trend in all the MILP classes we consider, there may still be challenging MILP classes where this linear trend does not hold. In other words, there may exist

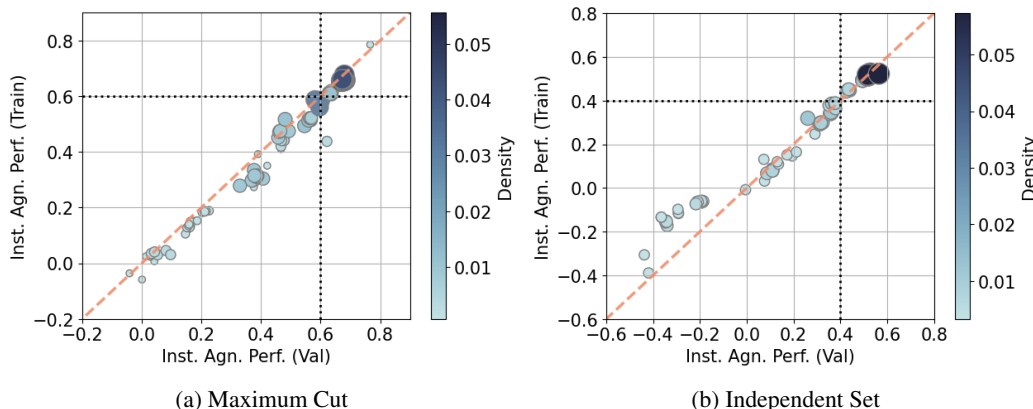

| (a) Maximum Cut | (b) Independent Set |

Figure 4: Instance-agnostic performance of configuration samples on the training set $\hat{\bar{\delta}}^{\mathcal{K}_{small}}(s)$ and hold-out set $\hat{\bar{\delta}}^{\mathcal{V}}(s)$ for Maximum Cut and Independent Set. The dashed orange line indicates the line of equality ($y = x$). The darkness and size of each circle (configuration) in the plot are proportional to the total number of configurations in the corresponding bin, divided by the number of samples selected from that bin. The horizontal and vertical black dotted lines indicate our choice of filtering threshold $b$ in Alg. 1. Respectively, 38% and 58% of all configurations in the initial configuration space $S$ surpass the filtering threshold and are considered as candidates for the final subspace $A$.

certain MILP classes where configurations that perform well on a training set may fail to generalize to the unseen test set. In such cases, we can modify our Alg. 1 to incorporate an additional holdout set $\mathcal{V}$, and filter configurations based on the performance on the hold out set $\hat{\bar{\delta}}^{\mathcal{V}}(s)$ instead of on the training set $\hat{\bar{\delta}}^{\mathcal{K}_{small}}(s)$ (See Line 16 in Alg. 1: $\hat{\bar{\delta}}(s) = \hat{\bar{\delta}}^{\mathcal{K}_{small}}(s)$ in our default algorithm, but can be replaced by $\hat{\bar{\delta}}^{\mathcal{V}}(s)$). In this way, we can more accurately capture the generalization behavior of each configuration, although this modification would increase the number of MILP solver calls required to collect the validation performances, which our reduction to solely monitor training performances $\hat{\bar{\delta}}^{\mathcal{K}_{small}}(s)$ avoids.

### A.2.3 Proof of Proposition 2

**Proposition 2. (Submodularity of $\hat{\Delta}(\tilde{\mathbf{f}}_A^{\mathbf{ERM}})$ and the greedy approximation algorithm).** The empirical performance of the ERM predictor $\hat{\Delta}(\tilde{f}_A^{ERM})$ is a monotone increasing and submodular function in $A$, and a greedy strategy where we include the configuration that achieves the greatest marginal improvement $\arg\max_{s \in \{0,1\}^M \setminus A} \hat{\Delta}(\tilde{f}_{A \cup \{s\}}^{ERM}) - \hat{\Delta}(\tilde{f}_A^{ERM})$ at each iteration is a $(1 - 1/e)$-approximation algorithm for constructing the subspace $A$ that optimizes $\hat{\Delta}(\tilde{f}_A^{ERM})$.

**Proof of monotonicity.** By definition, we have $\hat{\Delta}(\tilde{f}_A^{ERM}) = \frac{1}{K} \sum_{i=1}^{K} \delta(\tilde{f}_A^{ERM}(x_i), x_i)$ on a restricted subspace $A$. According to the ERM rule, for each instance $x_i \in \mathcal{M}$ we have $\delta(\tilde{f}_A^{ERM}(x_i), x_i) = \max_{s \in A} \delta(s, x)$. We note that $\delta(\tilde{f}_A^{ERM}(x_i), x_i)$ is a monotone increasing function in $A$ for each $x_i$, since if $B \subseteq C$, then $\delta(\tilde{f}_B^{ERM}(x_i), x_i) \leq \delta(\tilde{f}_C^{ERM}(x_i), x_i)$ due to the monotonicity of the max operator on the set. Averaging across all instances, $\hat{\Delta}(\tilde{f}_A^{ERM})$ is hence a monotone increasing function in $A$.

**Intuition for submodularity.** Adding a configuration $s$ to a set of configurations $A$ improves $\delta(\tilde{f}_{A \cup \{s\}}^{ERM}(x_i), x_i)$ from $\delta(\tilde{f}_A^{ERM}(x_i), x_i)$ (positive marginal improvement) if $s$ performs better than all configurations in $A$ on the MILP instance $x_i$. Intuitively speaking, with a larger subspace $A$, it is less likely for $s$ to improve the performance, because there are more competing choices in $A$ that make it more difficult for $s$ to perform the best. Hence, we get a smaller marginal improvement when adding a configuration $s$ to a larger set of configurations for each instance, therefore making the empirical performance $\hat{\Delta}(\tilde{\mathbf{f}}_A^{\mathbf{ERM}})$ averaged across all instances submodular in $A$. We provide a rigorous proof of submodularity below.

**Proof of submodularity.** Let $B \subseteq C \subseteq O = \{0,1\}^M$ and $s \in O \setminus C$. We want to show

$$\hat{\Delta}(\tilde{f}_{B \cup \{s\}}^{ERM}) - \hat{\Delta}(\tilde{f}_B^{ERM}) \geq \hat{\Delta}(\tilde{f}_{C \cup \{s\}}^{ERM}) - \hat{\Delta}(\tilde{f}_C^{ERM}) \tag{10}$$

We have

$$\hat{\Delta}(\tilde{f}_{B \cup \{s\}}^{ERM}) - \hat{\Delta}(\tilde{f}_B^{ERM}) = \frac{1}{K} \sum_{i=1}^{K} [\delta(\tilde{f}_{B \cup \{s\}}^{ERM}(x_i), x_i) - \delta(\tilde{f}_B^{ERM}(x_i), x_i)] \tag{11}$$

We can split the set $\mathcal{M} = \{x_1, ..., x_N\}$ into two nonoverlapping subsets $\mathcal{M}_0$ and $\mathcal{M}_1$ where

- $\forall x \in \mathcal{M}_0$, some configuration $s' \in B$ performs at least as good as $s$. That is, $\tilde{f}_{B \cup \{s\}}^{ERM}(x) = \arg\max_{s' \in B \cup \{s\}} \delta(s', x) \in B$, and hence

$$\delta(\tilde{f}_{B \cup \{s\}}^{ERM}(x), x) = \delta(\tilde{f}_B^{ERM}(x), x) \geq \delta(s, x) \tag{12}$$

- $\forall x \in \mathcal{M}_1$, the configuration $s$ performs better than all configurations in $B$. That is, $\tilde{f}_{B \cup \{s\}}^{ERM}(x) = \arg\max_{s' \in B \cup \{s\}} \delta(s', x) = s$, and hence

$$\delta(\tilde{f}_{B \cup \{s\}}^{ERM}(x), x) = \delta(s, x) > \delta(\tilde{f}_B^{ERM}(x), x) \tag{13}$$

Then, from Eq. (11), we have

$$\hat{\Delta}(\tilde{f}_{B \cup \{s\}}^{ERM}) - \hat{\Delta}(\tilde{f}_B^{ERM}) = \frac{1}{K} \sum_{x \in \mathcal{M}_1} [\delta(s, x) - \delta(\tilde{f}_B^{ERM}(x), x)] \tag{14}$$

Now consider

$$\hat{\Delta}(\tilde{f}_{C \cup \{s\}}^{ERM}) - \hat{\Delta}(\tilde{f}_C^{ERM}) = \frac{1}{K} \sum_{i=1}^{K} [\delta(\tilde{f}_{C \cup \{s\}}^{ERM}(x_i), x_i) - \delta(\tilde{f}_C^{ERM}(x_i), x_i)] \tag{15}$$

We have the following nonoverlapping cases

- $\forall x \in \mathcal{M}_0$, due to monotonicity of $\hat{\Delta}(\tilde{f}_A^{ERM})$ in $A$ and the fact that $B \subseteq C$, we have, extending from Eq. (12),

$$\delta(\tilde{f}_C^{ERM}(x), x) \geq \delta(\tilde{f}_B^{ERM}(x), x) \geq \delta(s, x) \tag{16}$$

and hence some configuration $s' \in C$ performs at least as good as $s$.

- $\forall x \in \mathcal{M}_1$, we further split into two nonoverlapping cases:

  (i) $s$ performs better than all configurations in $C$. That is, $\tilde{f}_{C \cup \{s\}}(x) = s$ and

$$\delta(\tilde{f}_{C \cup \{s\}}^{ERM}(x), x) = \delta(s, x) > \delta(\tilde{f}_C^{ERM}(x), x) \geq \delta(\tilde{f}_B^{ERM}(x), x) \tag{17}$$

  (ii) some configuration in $C \setminus B$ performs better than $s$. That is, $\tilde{f}_{C \cup \{s\}}^{ERM}(x) \in C \setminus B$ and

$$\delta(\tilde{f}_{C \cup \{s\}}^{ERM}(x), x) = \delta(\tilde{f}_C^{ERM}(x), x) \geq \delta(s, x) \geq \delta(\tilde{f}_B^{ERM}(x), x) \tag{18}$$

where the last inequality is from Eq. (13).

We let $\mathcal{M}_1 = \mathcal{M}_{11} \cup \mathcal{M}_{12}$ where $\mathcal{M}_{11}$ and $\mathcal{M}_{12}$ corresponds to (i) and (ii).

We thus have

$$
\begin{aligned}
\hat{\Delta}(\tilde{f}_{C \cup \{s\}}^{ERM}) - \hat{\Delta}(\tilde{f}_C^{ERM}) &= \frac{1}{K} \sum_{x \in \mathcal{M}_{11} \subseteq \mathcal{M}_1} [\delta(s, x) - \delta(\tilde{f}_C^{ERM}(x), x)] \\
&\leq \frac{1}{K} \sum_{x \in \mathcal{M}_{11} \subseteq \mathcal{M}_1} [\delta(s, x) - \delta(\tilde{f}_B^{ERM}(x), x)] \\
&\leq \frac{1}{K} \sum_{x \in \mathcal{M}_1} [\delta(s, x) - \delta(\tilde{f}_B^{ERM}(x), x)] \\
&= \hat{\Delta}(\tilde{f}_{B \cup \{s\}}^{ERM}) - \hat{\Delta}(\tilde{f}_B^{ERM})
\end{aligned}
\tag{19}
$$

where the second inequality is due to the monotonicity of $\delta(\tilde{f}_A^{ERM}(x), x)$ in $A$ for all $x$ (see Eq. (17)), the third inequality is due to nonnegativity of the additional terms in $\mathcal{M}_1 \setminus \mathcal{M}_{11}$ (see Eq. (18)), and the last equality is from Eq. (14).

**The greedy approximation algorithm.** Due to the monotone submodularity of the empirical performance of the ERM predictor $\hat{\Delta}(\tilde{f}_A^{ERM})$, the greedy strategy where we include the configuration that achieves the greatest marginal improvement $\arg\max_{s \in \{0,1\}^M \setminus A} \hat{\Delta}(\tilde{f}_{A \cup \{s\}}^{ERM}) - \hat{\Delta}(\tilde{f}_A^{ERM})$ at each iteration is a $(1 - 1/e)$-approximation algorithm for constructing the subspace $A$, as proven in previous work [41]. ∎

### A.2.4 ERM assumption discussion and relaxation to predictors with training error

Assuming that the predictor $\tilde{f}_A$ is the ERM predictor $\tilde{f}_A^{ERM}$ that performs optimally on the training set, we can construct a subspace $A$ prior to learning the actual predictor $\tilde{f}_A$ by replacing the learned predictor's prediction with the ERM selection rule. This assumption is reasonable because during training, we optimize the predictor with the empirical performance $\hat{\Delta}(\tilde{f}_A)$ as the objective, which aligns with the objective of the ERM predictor (where ERM obtains the optimal solution). To account for potential training errors, we can relax the ERM assumption with the following lemma, from which we achieve a trade-off similar to Eq. (3) of the main paper, balancing the ERM performance and a generalization term with a higher weight on the latter (see Eq. (24)). Our Alg. 1 hence still applies.

**Lemma 3.** Assume that a predictor $\tilde{f}_A$, when trained on $\mathcal{K}$, achieves optimal training performance (i.e., ERM $\tilde{f}_A^{ERM}$) with probability $1 - \beta$. With probability $\beta$, the predictor makes mistakes and outputs a configuration $s \in A$ uniformly at random. Then, combining with the assumption in Proposition 1 (See Appendix A.2.2), the trainset performance v.s. generalization decomposition can be written as

$$\Delta(\tilde{f}_A) = (1 - \alpha)(1 - \beta)\hat{\Delta}(\tilde{f}_A^{ERM}) + (1 - \alpha)\beta \frac{1}{|A|} \sum_{s \in A} \hat{\bar{\delta}}(s) + \alpha \frac{1}{|A|} \sum_{s \in A} \bar{\delta}(s) \tag{20}$$

**Proof.** By definition, $\hat{\Delta}(\tilde{f}_A) = \frac{1}{K}\sum_{i=1}^{K}\delta(f(x_i), x_i)$. Following the similar proof structure as Proposition 1, we have

$$\hat{\Delta}(\tilde{f}_A) = \begin{cases} \hat{\Delta}(\tilde{f}_A^{ERM}) & \text{with probability } \beta \\ \frac{1}{A}\sum_{s\in A}\frac{1}{K}\sum_{i=1}^{K}\delta(s, x_i) = \frac{1}{A}\sum_{s\in A}\hat{\bar{\delta}}(s) & \text{with probability } 1-\beta \end{cases} \tag{21}$$

Hence, we get

$$\hat{\Delta}(\tilde{f}_A) = (1-\beta)\hat{\Delta}(\tilde{f}_A^{ERM}) + \beta\frac{1}{A}\sum_{s\in A}\hat{\bar{\delta}}(s) \tag{22}$$

Before we further proceed in the proof, we discuss an additional trade-off when constructing the subspace $A$ based on the empirical performance on the training set $\hat{\Delta}(\tilde{f}_A)$. While adding more configurations to $A$ may improve the empirical performance of the ERM predictor $\tilde{f}_A^{ERM}$, some of these configurations may have low instance-agnostic performance and only perform well on a small subset of the training instances. Incorporating such configurations into $A$ may lead to the selection of poor configurations when training error occurs, resulting in a decrease in the performance on the training set $\hat{\Delta}(\tilde{f}_A)$. Hence, to construct a subspace $A$ that can result in the high empirical performance of the imperfect predictor, we also need to balance the size and diversity of $A$ (measured by the empirical performance of the ERM predictor on $A$, the first term), and the average configuration quality in $A$ (measured by the average instance-agnostic empirical performance, the second term).

Now, combining with the proof of Proposition 1, we hence have

$$\begin{aligned}\Delta(\tilde{f}_A) &= \hat{\Delta}(\tilde{f}_A) - \left(\hat{\Delta}(\tilde{f}_A) - \Delta(\tilde{f}_A)\right) \\ &= (1-\alpha)\hat{\Delta}(\tilde{f}_A) + \alpha\frac{1}{A}\sum_{s\in A}\bar{\delta}(s) \\ &= (1-\alpha)\left((1-\beta)\hat{\Delta}(\tilde{f}_A^{ERM}) + \beta\frac{1}{A}\sum_{s\in A}\hat{\bar{\delta}}(s)\right) + \alpha\frac{1}{A}\sum_{s\in A}\bar{\delta}(s) \\ &= (1-\alpha)(1-\beta)\hat{\Delta}(\tilde{f}_A^{ERM}) + (1-\alpha)\beta\frac{1}{A}\sum_{s\in A}\hat{\bar{\delta}}(s) + \alpha\frac{1}{A}\sum_{s\in A}\bar{\delta}(s) \quad\blacksquare\end{aligned} \tag{23}$$

Then, following Eq. (3) of the main paper, we replace $\bar{\delta}(s)$ (unobservable) by $\hat{\bar{\delta}}(s)$ and arrive at the following objective to select the subspace $A$:

$$\begin{aligned}\Delta(\tilde{f}_A) &= (1-\alpha)(1-\beta)\hat{\Delta}(\tilde{f}_A^{ERM}) + (\alpha + \beta - \alpha\beta)\frac{1}{|A|}\sum_{s\in A}\hat{\bar{\delta}}(s) \\ &= (1-\gamma)\hat{\Delta}(\tilde{f}_A^{ERM}) + \gamma\frac{1}{|A|}\sum_{s\in A}\hat{\bar{\delta}}(s) \quad\text{, where } \gamma = \alpha + \beta - \alpha\beta.\end{aligned} \tag{24}$$

Comparing the above with Eq. (3) where we assume the predictor performs ERM perfectly with no training error, we arrive at the same trade-off between the empirical performance of the ERM predictor $\tilde{f}_A^{ERM}$ in the subspace $A$ (which is monotone submodular in $A$), and the average instance-agnostic performance of all configurations $s \in A$, with a lower weight on the first term and a higher weight on the second term due to training error ($\gamma \geq \alpha$). Hence, our algorithm that couples a greedy strategy with the filtering criterion naturally applies to this relaxed scenario. The greedy strategy can still select configuration based on the ERM predictor, as the training error from the new predictor is absorbed in the second term. Due to the increase weight on the second term, we would increase the threshold $b$, which we design as a hyperparameter in Alg. 1, to more aggressively filter out configuration $s$ with a low instance-agnostic performance given by $\hat{\bar{\delta}}(s) \leq b$. We leave it as future work to analyze more complicated predictors $\tilde{f}'_A$ that incorporate other smoothness assumptions and to adapt the construction algorithm based on the performance of such predictors.

## A.3 Configuration Space Restriction: Algorithm

### A.3.1 Algorithm

The algorithm for our data-driven configuration space restriction in Sec. 5.1 is presented in Alg. 1.

---

**Algorithm 1:** Configuration_Space_Restriction

---

**Input:** MILP training set $\mathcal{K}_{small}$, the unrestricted configuration space $O = \{0,1\}^M$, number of initial configuration samples $|S|$, size of the restricted configuration space $|A|$, instance-agnostic performance threshold $b$

**Output:** The restricted configuration space $A$

1   $S \leftarrow$ large subset of $O$ by sampling $|S|$ configurations from $O$     // See description below
2   // construct the relative time improvement table
3   $T \leftarrow zeros(|S|, |\mathcal{K}_{small}|)$
4   **for** *Configuration $s_i$ in $S$* **do**
5      **for** *Instance $x_j$ in $\mathcal{K}_{small}$* **do**
6         // Solve instance $x_j$ with configuration $s_i$ with the MILP solver
7         $T_{ij} \leftarrow \delta(s_i, x_j)$
8      **end for**
9   **end for**
10   // construct the restricted configuration space
11   $A \leftarrow \{\}$
12   **for** *Choice i = 1: |A|* **do**
13      // greedy based on marginal improvement in instance-aware perf. (see Table 5)
14      $s \leftarrow \underset{s \in S \setminus A}{\arg\max} \ \hat{\Delta}(\tilde{f}_{A \cup \{s\}}^{ERM}, \mathcal{X}) - \hat{\Delta}(\tilde{f}_A^{ERM}, \mathcal{X})$
15      // filtering based on instance-agnostic perf. of individual configuration (see Table 5)
16      **if** $\hat{\tilde{\delta}}(s) > b$ **then**
17         $A \leftarrow A \cup \{x\}$
18      **else**
19         $S \leftarrow S \setminus \{x\}$
20      **end if**
21   **end for**

---

On Line 1, we use the following two strategies to sample the large initial configuration subset $S \subseteq O$:

1. **Near Zero:** we include all configurations that activate at most 3 separators, which results in a subset $S_1$ of size $\sum_{i=0}^{3} \binom{M}{i} = \sum_{i=0}^{3} \binom{17}{i} = 834$.

2. **Near Best Random:** we first sample 500 configurations uniformly at random from $O$ and find the configuration $\tilde{s}'$ in the sample with the highest empirical instance-agnostic performance $\hat{\tilde{\delta}}(s)$ on the training set $\mathcal{K}_{small}$ (see Table 5). Then, we include (1) all configurations that have at most 3 separator different from $\tilde{s}'$, resulting in a subset $S_{21}$ of size 834, and (2) all configurations whose set of activated separators is a subset of the activated separators in $\tilde{s}'$, resulting in another subset $S_{22}$ whose size depends on the number of activated separators in $\tilde{s}'$ and ranges from 63 to 1023 for all MILP classes considered in this paper.

Combining all samples from above, we obtain a large initial configuration subset $S$ with $|S| \approx 2000$.

The **Near Best Random** strategy is designed to bootstrap high quality samples around the high quality configuration $\tilde{s}'$ obtained from random search. The subset $S_{21}$ increases sample diversity by perturbing $\tilde{s}'$ within a Hamming distance of 3, and $S_{22}$ is designed based on the intuition that it may be possible to deactivate more separators from $\tilde{s}'$, as some activated separators in $\tilde{s}'$ may be useful for certain MILP instances but not for the others. The **Near Zero** strategy is designed based on the intuition that it may be beneficial to maintain a small set of activated separators, as it reduces the time to invoke separator algorithms (although at the cost of reducing the quality of the generated cuts).

### A.3.2 Algorithm discussions: filtering and subspace size

When employing our filtering strategy, a higher (more aggressive) threshold $\hat{\bar{\delta}}(s) \leq b$ with larger $b$ leads to a higher average empirical instance-agnostic performance (second term $\frac{1}{|A|} \sum_{s \in A} \hat{\delta}(s)$ in Eq. (3) of the main paper), which measures generalizability of the instance-aware predictor, but it also incurs a decrease in the empirical performance for the instance-aware ERM predictor (first term $\hat{\Delta}(\tilde{f}_A)$ in Eq. (3)), which measures the training performance of the instance-aware predictor).

In Figure 5, we plot the behavior of these two terms across different threshold values of $b$ (ignoring the weight of $\alpha$), using the $\mathcal{K}_{small}$ training set of Independent Set and Load Balancing. The size of the subspace $A$ is fixed at $|A| = 15$ for Independent Set and $|A| = 25$ for Load Balancing, which equals the size of our chosen subspace in L2Sep that is constructed with filtering threshold $b^{ours} = 0.4$ for Independent Set and $b^{ours} = -0.1$ for Load Balancing. The subspaces are constructed by our Alg. 1 that combines a greedy strategy with the filtering criterion.

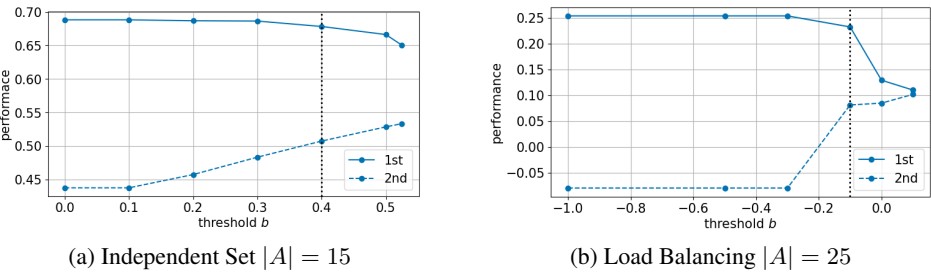

(a) Independent Set $|A| = 15$        (b) Load Balancing $|A| = 25$

Figure 5: Training set performance ($1^{st}$) and generalization ($2^{nd}$) terms in Eq. (3) of the main paper for subspaces constructed using Alg. 1 with varying thresholds $b$. The vertical dotted black lines indicate our chosen threshold.

An effective approach for choosing the threshold, as supported by our theoretical analysis in Sec. 5.1 of the main paper, is to find a value $b$ that yields a substantial improvement in the generalization term ($2^{nd}$), while simultaneously maintaining a high training set performance term ($1^{st}$). Our selection of $b^{ours} = 0.4$ for Independent Set and $b^{ours} = -0.1$ for Load Balancing satisfies this criterion.

In Figure 6, we further plot the two terms during the intermediate construction process of Alg. 1 where the subspace $A$ is expanded by adding a configuration at each iteration. The plot shows the behavior of the two terms through the construction process for a set of thresholds $b$ (as specified in the legend), when evaluated on the same $\mathcal{K}_{small}$ training dataset.

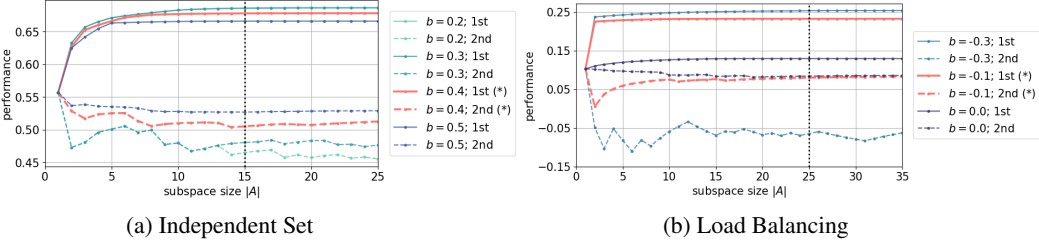

(a) Independent Set        (b) Load Balancing

Figure 6: Training set performance ($1^{st}$) and generalization ($2^{nd}$) terms in Eq. (3) for intermediate subspaces constructed using Alg.1 across a set of thresholds $b$. The vertical dotted black lines indicate our chosen subspace size. The asterisks (*) in the legend indicate our choice of $b$.

Once again, our choice of the threshold $b$ allows a notable improvement in the second generalization term ($2^{nd}$), while maintaining a high level of performance in the first training set performance term ($1^{st}$). This trend persists throughout each step of our iterative algorithm. We choose the size of the subspace $|A|$ (the termination criterion of our algorithm) when $|A|$ is reasonably small, while both terms stabilize and at values that offer a favorable trade-off between the two terms.

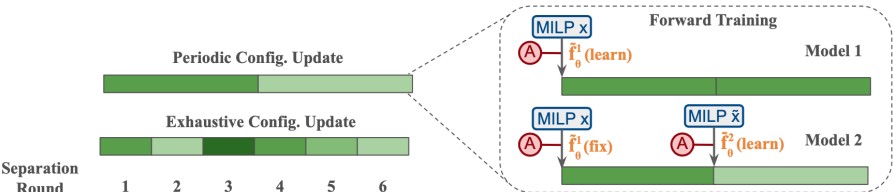

Figure 7: **Forward training.** (Left) The periodic update scheme where the configuration is updated at different separation rounds ($n_1 = 1$ and $n_2 = 4$ in the illustration) and held constant between updates (c.f. exhaustive update at each of the $R = 6$ rounds). (Right) The forward training algorithm that sequentially learn the $k$ networks $\{\tilde{f}_{\theta^T}^m\}_{m=1}^k$, whose output spaces are constrained to the subspace $A$, for periodic configuration update. The MILP input to $\tilde{f}_\theta^2$ is denoted as $\tilde{x}$ as it includes both the initial MILP $x$ and newly added cuts. Different shades of green represent different selected configurations.

## A.4   Configuration Update Restriction

### A.4.1   Forward training algorithm

Alg. 2 presents our forward training procedure that trains $k$ predictor networks to perform $k$ configuration updates for each MILP instance. As illustrated in Fig. 7, when training the $j^{th}$ network, we freeze the weights of the pre-trained networks $\{\tilde{f}_{\theta^T}^m\}_{m=1}^{j-1}$ and use them to update the configurations at separation rounds $\{n_m\}_{m=1}^{j-1}$. Then, we learn the $j^{th}$ network $\tilde{f}_{\theta^0}^j$ using neural UCB to update the configuration at separation round $n_j$, and we hold the configuration constant until the solver terminates (at optimality or a fixed gap) to collect the terminal reward. We do not use intermediate rewards as both the optimality gap and solve time vary at an intermediate round, making it difficult to construct an integrated reward to compare different configurations.

---

**Algorithm 2:** Forward_Training

**Input:** MILP training set $\mathcal{K}_{large}$, number of configuration update steps $k$, configuration predictor networks $\{f_{\theta^0}^j\}_{j=1}^k$, separation rounds $\{n_j\}_{j=1}^k$, training epochs $T$, number of MILP instances $P$ per epoch, number of samples to collect reward labels per epoch $D$, UCB scaling factor $\gamma$, configuration subspace $A$

**Output:** Trained regression networks $\{f_{\theta^T}^j\}_{j=1}^k$

1 **for** *Update j = 1: k* **do**
2     Freeze the weight of $\{f_{\theta^T}^m\}_{m=1}^{j-1}$
3     $f_{\theta^T}^j \leftarrow$ Neural_UCB $(\mathcal{K}_{large}, f_{\theta^0}^j, \{f_{\theta^T}^m\}_{m=1}^{j-1}, \{n_m\}_{m=1}^j, T, P, D, \gamma, A)$
4 **end for**

---

### A.4.2   Trade-off discussion for different k's

Let $\pi^{*(R)}$ and $\pi^{*(k)}$ be the optimal configuration policies when we perform $R$ and $k \ll R$ updates, and let $\tilde{\pi}^{(R)}$ and $\tilde{\pi}^{(k)}$ be the corresponding learned policies. Due to cascading errors over the long horizon, the learning task for $\tilde{\pi}^{(R)}$ is more challenging than for $\tilde{\pi}^{(k)}$; on the other hand, the optimal policy $\pi^{*(k)}$ performs worse than $\pi^{*(R)}$ due to the action space restriction. Hence, the frequency $k$ trades off approximation (for $\pi^{*(k)} \approx \pi^{*(R)}$) and estimation (for $\tilde{\pi}^{(k)} \approx \pi^{*(k)}$), with more frequent updates (larger $k$) improves the approximation error while less frequent updates (smaller $k$) improves the estimation error.

Recent theoretical work by Metelli et al. [38] investigates the impact of action persistence, namely repeating an action for a fixed number of decision steps, for infinite horizon discounted MDPs. They provide a theoretical bound on the approximation error in terms of the differences in the optimal Q-value with and without action persistence (which corresponds to the Q-value of $\pi^{*(R)}$ and $\pi^{*(k)}$ in our setting). The resulting bound is a function of the discount factor, action hold length, and the discrepancy between the transition kernel with and without action persistence. The approximation error is agnostic to the specific learning algorithm, and hence the analysis can be adapted to our

setting by extending it to the finite horizon MPD scenario. They further use fitted Q-iteration to learn the policies and establish a theoretical bound on the estimation error in terms of the differences in the Q-value of the learned policies with and without action persistence. In contrast, our learned policies $\tilde{\pi}^{(R)}$ and $\tilde{\pi}^{(k)}$ are trained via the forward training algorithm. While it is not the focus of our paper, we note a possible future research to extend their theoretical analysis of the estimation error to the forward training algorithm, and compare the theoretical bounds on approximation and estimation to analyze the trade-off associated with the configuration update frequency $k$.

### A.5 Neural UCB Algorithm

#### A.5.1 Training algorithm

The Neural UCB algorithm [59] that we employ to train each configuration predictor network is presented in Alg. 3.

---

**Algorithm 3:** Neural_UCB

---

**Input:** MILP training set $\mathcal{K}_{large}$, predictor network at the current separation round $\tilde{f}_{\theta^0}^j$, trained predictor networks at previous separation rounds $\{\tilde{f}_{\theta^T}^m\}_{m=1}^{j-1}$, current separation round $n_j$, previous separation rounds $\{n_m\}_{m=1}^{j-1}$, training epochs $T$, number of MILP instances $P$ per epoch, number of samples to collect reward labels per epoch $D$, UCB scaling factor $\gamma$, UCB regularization parameter $\lambda$, configuration subspace $A$

**Output:** Trained predictor network $\tilde{f}_{\theta^T}^j$

1 Initialize $B^0 \leftarrow$ an empty training data buffer
2 Initialize $Z_0 \leftarrow \lambda I_{|\theta| \times |\theta|}$
3 **for** *Epoch t = 1: T* **do**
4      Initialize $B^t \leftarrow B^{t-1}$
5      **for** *P iterations* **do**
6          Sample instance $x$ from $\mathcal{K}_{large}$
7          // sampling $D$ configurations with UCB to balance exploration and exploitation
8          **for** *Configuration s in A* **do**
9              Compute $U_{t,s} \leftarrow \tilde{f}_{\theta^{t-1}}^j(x, s_i) + \gamma \sqrt{\nabla_\theta \tilde{f}_{\theta^{t-1}}^j(x, s)^\mathsf{T} Z_{t-1}^{-1} \nabla_\theta \tilde{f}_{\theta^{t-1}}^j(x, s)}$
10          **end for**
11          Let $S \leftarrow D$ samples without replacement $\sim softmax_{s \in A}(U_{t,s})$
12          Compute $Z_t \leftarrow Z_{t-1} + \sum\limits_{s \text{ in } S} \nabla_\theta \tilde{f}_{\theta^{t-1}}^j(x, s) \nabla_\theta \tilde{f}_{\theta^{t-1}}^j(x, s)^\mathsf{T}$
13          // Collect the reward label for each configuration
14          **for** *Sample i = 1: D* **do**
15              Regression Label $r_i \leftarrow$ Run the MILP solver for the instance $x$: use $\{\tilde{f}_{\theta^T}^m\}_{m=1}^{j-1}$ to update configurations at separation rounds $\{n_m\}_{m=1}^{j-1}$, and update configuration to $s_i$ at separation round $n_j$
16          **end for**
17          $B^t \leftarrow B^t \cup \{x, s_i, r_i\}_{i=1}^{H+1}$
18      **end for**
19      $\tilde{f}_{\theta^t}^j \leftarrow$ Train $\tilde{f}_{\theta^{t-1}}^j$ with the updated buffer $B^t$
20 **end for**

---

#### A.5.2 Input features

Paulus et al. [42] design a comprehensive set of input features for variable and constraint nodes (extended from Gasse et al. [18] for their cut selection task), resulting in $\mathbf{V} \in \mathbb{R}^{n \times 17}$ and $\mathbf{C} \in \mathbb{R}^{m \times 34}$, where $m$ and $n$ are the number of constraints and variables in the MILP instance $x$. We note that the constraint nodes include the initial constraints from the MILP $x$ as well as the newly added cuts. We adopt their input features and provide a detailed description of the features in Table 6 for completeness of the paper. Meanwhile, we set the separator nodes (unique to our task) as $\mathbf{S} \in \mathbb{R}^{M \times 1}$,

where $M$ is the number of separators in the MILP solver, and each separator node $\mathbf{S}_k$ has a single dimensional binary feature indicating whether the separator is activated (1) or deactivated (0).

For edge weights, we similarly follow Paulus et al. [42] and Gasse et al. [18] to connect each variable-constraint node pair if the variable appears in a constraint, and set the edge weight to be the corresponding nonzero coefficient. Meanwhile, we connect all separator-variable and separator-constraint node pairs with a weight of 1, which results in a complete pairwise message passing between each separator-variable and separator-constraint pair in the Graph Convolution Network [33]. As we lack reliable prior knowledge of the weight for the separator-variable and separator-constraint pair, we do not provide initial weight information and instead directly use the graph convolution mechanism to automatically learn the similarity between each pair.

Table 6: Description of input features for variable and constraint nodes [42].

| Node Type | Feature | Description |
|---|---|---|
| **Vars** | norm coef | Objective coefficient, normalized by objective norm |
| | type | Type (binary, integer, impl. integer, continuous) one-hot |
| | has lb | Lower bound indicator |
| | has ub | Upper bound indicator |
| | norm redcost | Reduced cost, normalized by objective norm |
| | solval | Solution value |
| | solfrac | Solution value fractionality |
| | sol_is_at_lb | Solution value equals lower bound |
| | sol_is_at_ub | Solution value equals upper bound |
| | norm_age | LP age, normalized by total number of solved LPs |
| | basestat | Simplex basis status (lower, basic, upper, zero) one-hot |
| **Cons, Added Cuts** | is_cut | Indicator to differentiate cut vs. constraint |
| | type | Separator type, one-hot |
| | rank | Rank of a row |
| | norm_nnzrs | Fraction of nonzero entries |
| | bias | Unshifted side normalized by row norm |
| | row_is_at_lhs | Row value equals left hand side |
| | row_is_at_rhs | Row value equals right hand side |
| | dualsol | Dual LP solution of a row, normalized by row and objective norm |
| | basestat | Basis status of a row in the LP solution, one-hot |
| | norm_age | Age of row, normalized by total number of solved LPs |
| | norm_nlp_creation | LPs since the row has been created, normalized |
| | norm_intcols | Fraction of integral columns in the row |
| | is_integral | Activity of the row is always integral in a feasible solution |
| | is_removable | Row is removable from the LP |
| | is_in_lp | Row is member of current LP |
| | violation | Violation score of a row |
| | rel_violation | Relative violation score of a row |
| | obj_par | Objective parallelism score of a row |
| | exp_improv | Expected improvement score of a row |
| | supp_score | Support score of a row |
| | int_support | Integral support score of a row |
| | scip_score | SCIP score of a row for cut selection |

### A.6 Experiment Setups

#### A.6.1 Proposed method details

Our main result tables (Table 1 and Table 3 of the main paper) present our complete method and two sub-components. We provide details on the implementation of these three methods.

**(1) Ours (L2Sep)**: for each MILP class, we first run Alg. 1 to obtain a configuration subspace $A$. Then, we run Alg. 2 that uses forward training to learn $k = 2$ separate instance-aware configuration predictors $\tilde{f}_\theta^1$ and $\tilde{f}_\theta^2$ to perform two configuration updates at separation rounds $n_1$ and $n_2$. The output of the predictors is restricted to the subspace $A$. We train the predictors using the Neural UCB Algorithm 3. A summary of our training pipeline is shown in Alg. 4. At inference time, we have two selection strategies: we select the configuration with either the highest predicted reward $s_{x,j}^* = \arg\max_{s \in A} \tilde{f}_{\theta^T}^j(x,s)$ or the highest UCB score $s_{x,j}^* = \arg\max_{s \in A} ucb(x,s)$ for all predictors $j$, determined based on validation performance. In our experience, we observe that selecting configurations based solely on the point estimate $\tilde{f}_{\theta^T}^j(x,s)$ alone can sometimes be overly deterministic, leading to a limited range of configurations being chosen for most instances. The UCB score combines the reward point estimate $\tilde{f}_{\theta^T}^j(x,s)$ with an uncertainty estimate (computed in Line 8 of Alg. 3, where the normalizing matrix $Z$ is from the same epoch as the selected model), and it hence increases the diversity of the prediction and improves the performance in most scenarios. Given the selection strategy, for each instance $x$, we set the configuration to be $s_{x,1}^*$ at separation rounds $n_1$ and $s_{x,2}^*$ at separation rounds $n_2$. We hold the configuration fixed as $s_{x,1}^*$ between separation rounds $[n_1, n_2]$ and as $s_{x,2}^*$ from separation round $n_2$ until the solving process terminates.

---

**Algorithm 4:** Learning_To_Separate (L2Sep)

**Input:** MILP training set $\mathcal{K}_{small}$, the unrestricted configuration space $O = \{0,1\}^M$, number of initial configuration samples $|S|$, size of the restricted configuration space $|A|$, instance-agnostic performance threshold $b$, MILP training set $\mathcal{K}_{large}$, number of configuration update steps $k$, configuration predictor networks $\{f_{\theta^0}^j\}_{j=1}^k$, separation rounds $\{n_j\}_{j=1}^k$, training epochs $T$, number of MILP instances $P$ per epoch, number of samples to collect reward labels per epoch $D$, UCB scaling factor $\gamma$, UCB regularization parameter $\lambda$

**Output:** Trained regression networks $\{f_{\theta^T}^j\}_{j=1}^k$

1   $A \leftarrow$ Configuration_Space_Restriction $(\mathcal{K}_{small}, O, |S|, |A|, b)$   // See Alg. 1

2   $\{f_{\theta^T}^j\}_{j=1}^k \leftarrow$ Forward_Training $(\mathcal{K}_{large}, k, \{f_{\theta^0}^j\}_{j=1}^k, \{n_j\}_{j=1}^k, T, P, D, \gamma, \lambda, A)$   // See Alg. 2

---

**(2) Instance Agnostic Configuration**: The first step of Alg. 1 is to sample a large subspace $S$ with $|S| \approx 2000$ due to computational infeasiblity to enumerate all $\{0,1\}^M$ configurations (See Appendix A.3.1). For each configuration $s \in S$, we compute its empirical instance-agnostic performance $\hat{\bar{\delta}}(s) = \frac{1}{|\mathcal{K}_{small}|} \sum_{x \in \mathcal{K}_{small}} \delta(s, x)$ and choose the best (ERM) configuration $\tilde{s}_S = \arg\max_{s \in S} \hat{\bar{\delta}}(s)$ as the instance agnostic configuration. We apply the same $\tilde{s}_S$ to all instances in the given MILP class to evaluate its performance. Notably, $\tilde{s}_S$ always appears in the final restricted subspace $A$, as it is selected at the first iteration of Alg. 1 when the marginal improvement of the instance-aware function (Line 14) and the instance-agnostic performance (Line 16) coincide.

**(3) Random Within Restricted Subspace**: Given the subspace $A$ constructed by Alg. 1, we choose a configuration in the subspace $A$ uniformly at random for each MILP instance.

#### A.6.2 Parameters

Table 7 and Table 8 present a list of parameters along with their experimental values used by our L2Sep method in Alg. 4.

#### A.6.3 SCIP interface.

We use a custom version of the SCIP solver (v7.0.2) [8] provided by Paulus et al. [42] and a custom version of the PySCIPOpt interface (v3.3.0) [37] to add a special separator invoked at the beginning

Table 7: A list of parameters and their values as used in the experiments for our data-driven subspace restriction Alg. 1.

| **Tang** | Bin. Pack. | Max. Cut | Pack. |
|---|---|---|---|
| Number of initial config. samples $|S|$ | 1795 | 1795 | 2691 |
| Size of restricted config. space $|A|$ | 30 | 20 | 15 |
| Filtering threshold $b$ | 0.3 | 0.6 | 0.0 |
| **Ecole** | Comb. Auc | Indep. Set | Fac. Loc. |
| Number of initial config. samples $|S|$ | 1699 | 1699 | 1923 |
| Size of restricted config. space $|A|$ | 25 | 15 | 20 |
| Filtering threshold $b$ | 0.5 | 0.4 | 0.0 |
| **Real-world** | MIPLIB | NNV | Load Bal. |
| Number of initial config. samples $|S|$ | 2179 | 2691 | 1795 |
| Size of restricted config. space $|A|$ | 20 | 20 | 25 |
| Filtering threshold $b$ | -0.17 | 0.0 | -0.1 |

Table 8: A list of parameters and their values as used in the experiments for our learning Alg. 2 and 3.

| Parameter | Value | |
|---|---|---|
| Number of separators $M$ | 17 | |
| Number of configuration updates $k$ | 2 | |
| Configuration update round $n_1$ | 0 | |
| Configuration update round $n_2$ | 5 | for Tang instances |
| | 8 | for Ecole instances |
| | 10 | for MIPLIB, NNV, Load Balancing |
| Training epoch $T$ | 70 | |
| Number of MILP instances per epoch $P$ | 6 | |
| Number of samples to collect reward labels per epoch $D$ | 8 | |
| Number of solver runs to collect a reward label for each config.-instance pair $l$ | 3 | |
| Reward clipping constant $r_{\min}$ | -1.5 | |
| UCB scaling factor $\gamma$ | 0.9375 | |
| UCB regularization parameter $\lambda$ | 0.001 | |

of each separation round to activate and deactivate $M = 17$ standard separators implemented in SCIP, including *aggregation, cgmip, clique, cmir, convexproj, disjunctive, eccuts, flowcover, gauage, gomory, impliedbounds, intobj, mcf, oddcycle, rapidlearning, strongcg, zerohalf*[2]

### A.6.4 Training and evaluation details

**Architecture and training hyperparameters.** Our network first embeds $\mathbf{V} \in \mathbb{R}^{n \times 17}$, $\mathbf{C} \in \mathbb{R}^{m \times 34}$, and $\mathbf{S} \in \mathbb{R}^{M \times 1}$ (where $n, m, M$ are the number of variables, constraints, and separators) into hidden representations of dimension $d_{hidden} = 64$ with a BatchNorm followed by two (Linear, ReLU) blocks. Then, our Graph Convolution module [33] takes the hidden embeddings for message passing, following the direction of ($\mathbf{V}{\rightarrow}\mathbf{C}{\rightarrow}\mathbf{V}$, $\mathbf{S}{\rightarrow}\mathbf{V}{\rightarrow}\mathbf{S}$, and $\mathbf{S}{\rightarrow}\mathbf{C}{\rightarrow}\mathbf{S}$), with a final (LayerNorm, ReLU, Linear) block that maintains the dimension $d_{hidden}$. Then, the separator nodes $\mathbf{S}$ pass through a TransformerConv attention module [48] with $n_{heads} = 4$ heads and a *dropout* rate of 0.1. Lastly, we perform a global mean pooling on each of the $\mathbf{C}$, $\mathbf{V}$, and $\mathbf{S}$ hidden embeddings to obtain three embedding vectors, concatenate them into a single vector, and finally use a (Linear, ReLU, Linear) block to map the vector into a scalar output. We train with Adam optimizer with a learning rate of 0.001 and a batch size of 64 for 70 epochs with a total of 180000 gradient steps. All hyperparameters are selected on the validation set and frozen before evaluating on the test set. Table 9 and 10 provides a list of hyperparameters.

---

[2]Detailed descriptions of the separators can be found in `https://www.scipopt.org/doc-7.0.2/html/group__SEPARATORS.php`.

Table 9: **Architecture hyperparameters.**

| | | | |
|---|---|---|---|
| Input dimension $n \times d_n$ $m \times d_n$ $s \times d_s$ | $\mathbf{V} \in \mathbb{R}^{n \times 17}$ $\mathbf{C} \in \mathbb{R}^{m \times 34}$ $\mathbf{S} \in \mathbb{R}^{M \times 1}$ | GCN Message Passing Order | $\mathbf{V}\to\mathbf{C}\to\mathbf{V}$, $\mathbf{S}\to\mathbf{V}\to\mathbf{S}$, $\mathbf{S}\to\mathbf{C}\to\mathbf{S}$ |
| Output dimension | 1 | Attention Num. Heads | 4 |
| Embedding dimension $d_{hidden}$ | 64 | Attention Dropout | 0.1 |
| | | Activation | ReLU |

Table 10: **Training hyperparameters.**

| | |
|---|---|
| Optimizer | Adam |
| Learning rate | 0.001 |
| Batch size | 64 |
| Num. of Gradient Steps | 180000 |

**Data split, MILP solver termination criterion, and inference strategy.** For all MILP classes except MIPLIB, we collect a small training set $\mathcal{K}_{small}$ of 100 instances for configuration space restriction, and a large training set $\mathcal{K}_{large}$ of 800 instances for predictor network training. We hold out a validation set and a test set of 100 instances each. For MIPLIB, our curated subset contains 443 instances in total. We split the instances into $\mathcal{K}_{small}$ with 30 instances, $\mathcal{K}_{large}$ with 270 instances, a validation set with 55 instances, and a test set with 88 instances.

For all MILP classes except MIPLIB and Load Balancing, we solve the instances until optimality. For MIPLIB and Load Balancing, we solve all instances until it reaches a primal-dual gap of 10%.

For Packing and Bin Packing, our best inference selection strategy, chosen based on validation performance, is to select the configuration with the highest predicted reward $s^*_{x,j} = \arg\max_{s \in A} \tilde{f}^j_\theta(x, s)$. For all other MILP classes, the best inference selection strategy is to select the configuration with the highest UCB score $s^*_{x,j} = \arg\max_{s \in A} ucb(x, s)$.

**Training and evaluation.** We collect data, train, validate and test all methods on a distributed compute cluster using nodes equipped with 48 Intel AVX512 CPUs. A single Nvidia Volta V100 GPU is used to train all MILP classes except MIPLIB, as the massive number of variables and constraints (up to $1.4 \times 10^6$) in certain MIPLIB instances poses memory challenges for the GPU device (with 16GB memory). Our configuration subspace restriction Alg. 1 requires approximately 36 hours, while the training time for Alg. 2 is within 48 hours for all benchmarks including MIPLIB. In certain cases, the baseline methods take excessively long solve time, making it computationally expensive to evaluate those methods. During test evaluation, we terminate the solve if the solve time exceeds three times the SCIP default (which can be up to 20 minutes), resulting in a time improvement as low as $-300\%$. This time limit is more relaxed than the reward clipping $r_{\min} = -1.5$ during training (which corresponds to $\leq -150\%$ time improvement and is applied to improve data collection efficiency). Notably, the hard stop is not used for our complete learning method (L2Sep), whose performance improvement from SCIP default remains consistently stable across instances. Instead, it is primarily used for the random or prune baseline that exhibits very poor performance or a significantly large standard deviation. Without the time limit, these baselines may exhibit worse performance than those reported in Table 1 and 3 of the main paper.

### A.6.5 MILP benchmarks

We perform extensive evaluation on the following MILP benchmarks of a variety of different sizes.

**Standard: Tang et al.** We consider three out of four MILP classes introduced in Tang et al. [51]: Maximum Cut, Packing, and Binary Packing. We follow Paulus et al. [42] to only consider the large size where the number of variables and constraints $n, m \in [50, 150]$, as Paulus et al. [42] observe the small and medium size are too easy for the SCIP solver. We do not consider the Planning class because the large size is still too small, as SCIP takes less than $0.01s$ on average to solve the instances.

We generate large size Tang instances with class specific parameters $n = 60$ variables, $m_{\text{resource}} = 60$ constraints for Packing, $n = 66$ variables, $m_{\text{resource}} = 132$ constraints for Binary Packing, and $n_{\text{vertices}} = 54, n_{\text{edges}} = 134$ for Maximum Cut.

**Standard: Ecole.** We consider three out of four classes of instances from [43]: Combinatorial Auction, Independent Set, and Capacitated Facility Location, where the number of variables and constraints in the instances $n, m \in [100, 10000]$.

We generate Ecole instances with class specific parameters $n_{\text{items}} = 100, n_{\text{bids}} = 500$ for Combinatorial Auction, $n_{\text{nodes}} = 500$ for Independent Set, $n_{\text{rows}} = 500, n_{\text{columns}} = 1000$ for Set Cover, and $n_{\text{customers}} = 100, n_{\text{facilities}} = 100$ for Capacitated Facility Location.

When we use the default parameters in the Ecole library to generate the Combinatorial Auction and Independent Set datasets[3], we notice that the instance-aware ERM (optimal) predictor sometimes only selects a small number of configurations within each class, resulting in similar performance as the instance-agnostic configuration (single configuration) due to selection homogeneity. Despite both resulting in significant relative time improvements from SCIP default, the default parameters cause the instances within each of these two classes too similar to one another, thereby limiting the benefits of instance-aware configurations.

Table 11: Sampling parameter distributions for each Independent Set and Combinatorial Auction instance.

| Independent Set | | |
| --- | --- | --- |
| graph_type | edge_probability | affinity |
| $U(\{\text{barabasi\_albert}, \text{erdos\_renyi}\})$ | $U([0.005, 0.01])$ | $U(\{2, 3, 4, 5, 6\})$ |
| **Combinatorial Auction** | | |
| value_deviation | add_item_prob | max_n_sub_bids |
| $U([0.25, 0.75])$ | $U([0.5, 0.75])$ | $U(\{3, 4, 5, 6, 7\})$ |
| additivity | budget_factor | resale_factor |
| $U([-0.1, 0.4])$ | $U([1.25, 1.75])$ | $U([0.35, 0.65])$ |

To enhance instance diversity, we adjust the parameters for both classes by sampling uniformly at random from a broad distribution, as presented in Table 11, for each MILP instance. As many large-scale real-world MILP problems exhibit significant dataset heterogeneity (for instance, see MIPLIP below), we believe that our adjusted Ecole datasets better reflect realistic MILP scenarios.

We do not report results for Set Cover as we find an instance-agnostic configuration $s^{ERM}$ that deactivates *all* separators is able to achieve a relative time improvement of $88.6\%$ (IQM: $88.4\%$, mean: $87.4\%$, standard deviation $6\%$) from SCIP default, whereas the default median solve time is $5.96s$ (IQM $6.13s$, mean: $6.45s$, standard deviation $3.17s$). Furthermore, on the small training set $\mathcal{K}_{small}$, we find that the instance-aware empirical performance of the ERM predictor $\tilde{f}^{ERM}$ differs by less than $1\%$ from the performance of $s^{ERM}$. We experiment with various parameters for the Set Cover problem, such as the number of rows and columns, density of the constraint matrix, and maximum objective coefficient, but observe similar outcomes. This implies that separators in the SCIP solver do not provide significant benefits for the Set Cover class, and therefore, the instance-aware ERM predictor aligns with the instance-agnostic configuration to deactivate all separators.

**Large-scale: NN Verification.** We consider the large-scale neural network verification instances used in Paulus et al. [42], with a median size $(n, m)$ of the number of variables and constraints at $(7142, 6531)$. This dataset formulates MILPs to verify whether a convolutional neural network is robust to input perturbations on each image in the MNIST dataset. The MILP formulation of the

---

[3]`https://doc.ecole.ai/py/en/stable/reference/instances.html` provides a list of adjustable parameters for each Ecole MILP class.

verification problem can be found in Gowal et al. [21]. We follow Paulus et al. [42] to exclude all infeasible instances (often trivially solved at presolve) and instances that reach a 1-hour time-limit in SCIP default mode. Due to differences in the learning tasks (separator configuration for the entire solve v.s. cut selection at the root node), we further exclude instances that cannot be solved optimally within 120 seconds, after which 74% of the instances used in Paulus et al. [42] remain.

**Large-scale: MIPLIB.** MIPLIB2017 collection [20] is a large-scale *heterogeneous* MILP benchmark dataset that contains a curated set of challenging real-world instances from various application domains. It contains 1065 instances where the number of variables and constraints $(n, m)$ varies from 30 to $1,429,098$. Previous work [52] finds it very challenging to learn over MIPLIB due to dataset heterogeneity. A recent work [54] attempts to learn cutting plane selection over two subsets of similar instances (with 20 and 40 instances each), curated via instance clustering from two starting instances containing knapsack and set cover constraints.

We attempt to learn over a larger heterogeneity subset of the MIPLIB dataset to solve the separator configuration task. We adopt a similar dataset pre-filtering procedure as in Turner et al. [52], where we discard instances that are infeasible, solved after presolving, or the primal-dual gaps are larger than 10% after 300 seconds of solve time. It is worth noting that our pre-filtering procedure preserves the dataset heterogeneity, whereas Wang et al. [54] reduces the dataset to homogeneous subsets. We obtain a subset of 443 instances, which is around 40% of the original MIPLIB dataset.

**Large-scale: Load Balancing (ML4CO Challenge).** We consider the server load balancing dataset from Neurips 2021 ML4CO Challenge [19] [4], whose average number of variables and constraints $(n, m)$ is at $(64304, 61000)$. This dataset is inspired by real-world applications in distributed computing, where the goal is to allocate data workloads to the fewest number of servers possible while ensuring that the allocation remains resilient to the failure of any worker. This is a challenging dataset due to the large number of variables and constraints, along with the presence of nonstandard robust apportionment constraints that separators in MILP solvers may not be specialized for. No instances from the original load balancing dataset are pre-filtered, as we observe that the primal-dual gaps for all instances remain within 10% after 300 seconds of solve time.

The ML4CO challenge releases two other datasets: Item Placement and Anonymous. We follow Wang et al. [54] and exclude Item Placement since very few cutting planes are generated and used in the dataset (with an average of less than five candidate cuts on each instance), thereby limiting the influence of separators on the dataset. Additionally, we exclude Anonymous because the dataset is restrictively small for effective learning (comprising only 98 train and 20 valid instances).

---

[4]More details on the dataset can be found at `https://www.ecole.ai/2021/ml4co-competition/`

## A.7 Ablation Details

### A.7.1 Implementation details of ablation methods

**Configuration space restriction (Sec. 5.1):**

**(1) No Restr.**: We do not constrain the output space of the $k = 2$ instance-aware predictors, allowing them to output any configuration in the unrestricted space $\{0, 1\}^M$. Due to the vast number of actions, it is infeasible to use the neural UCB Alg. 3 as the normalizing matrix $Z$ becomes excessively large, making it challenging to obtain meaningful confidence bounds as most actions remain unexplored. Hence, we use the following $\epsilon$-greedy exploration strategy (similar to the $\epsilon$-**greedy** ablation below): we first sample a subset of $50$ configurations for training efficiency. Then, we iteratively select $D$ configurations from the subset to collect the reward labels: at each of the $D$ iteration, with probability $\epsilon = 0.1$ we select a configuration uniformly at random among the unselected ones in the subset, and with probability $1 - \epsilon$ we choose the configuration with the highest reward point estimate among the unselected ones in the subset. At inference time, we sample a subset of $500$ configurations for each MILP instance, and select the configuration with the highest reward point estimate. All the other settings remain the same as our complete method (L2Sep). This ablation is used to examine the effectiveness of restricting configuration space in learning better configuration predictors.

**(2) Greedy Restr.**:
When we run Alg. 2 to obtain the configuration subspace $A'$, we select configurations solely with the greedy strategy based on the highest marginal improvement, but we do not filter out configurations with low instance-agnostic performance (equivalent, we set the filtering threshold $b = -\infty$). The rest of the learning remains identical to our complete method (L2Sep), but with the predictor's output restricted to the greedy subspace $A'$. This ablation allows us to assess the effectiveness of the filtering strategy in constructing a superior configuration subspace for learning predictors.

**Configuration step restriction (Sec. 5.2):**

**(1) k = 1**: We perform one configuration update for each MILP instance. We use the first config. predictor $\tilde{f}_\theta^1$ from our complete method (L2Sep) to set configuration at separation round $n_1$.

**(2) k = 3**: We fix the two configuration predictors $\tilde{f}_\theta^1$ and $\tilde{f}_\theta^2$ from our complete method (L2Sep), and follow Alg. 2 to train a third configuration predictor $\tilde{f}_\theta^3$ at separation round $n_3$ ($= 12$ for Tang instances and 20 for Ecole instances). The predictor $\tilde{f}_\theta^3$ is similarly restricted to the subspace $A$. At test time, for each instance $x$, we follow our complete method to perform two configuration updates using $\tilde{f}_\theta^1$ and $\tilde{f}_\theta^2$ until separation round $n_3$. Then, we use $\tilde{f}_\theta^3$ to update the configuration at separation round $n_3$ and hold it fixed until the solving process terminates.

**Neural UCB Algorithm (Sec. 5.3):**

**(1) Supervise ($\times 4$)**: We exactly follow our complete method (L2Sep), except that we use offline regression instead of online neural UCB to train the predictors. Offline regression first collects a large training set of instance-configuration-reward tuples offline (by randomly sampling instance-configuration pairs and using the MILP solver to obtain the reward labels). The network is then trained on the fixed training set. In contrast, neural UCB gradually expands the training buffer by collecting training data online, while using the current trained model to guide exploration (sampling configurations with high uncertainty) and exploitation (sampling configurations with high predicted reward). Model training is performed online on the continually updated dataset. Due to the difference in online v.s. offline nature of the dataset generation and training scheme, one learning method may require more gradient updates to converge than the other. To attempt at a fair comparison, we collect $\times 4$ more instance-configuration-reward tuples and train the offline network until convergence. This ablation is used to examine the efficacy of online learning through neural contextual bandit in improving the training efficiency of predictor networks.

**(2) $\epsilon$-greedy**: We exactly follow our complete method, except when training the predictors with the Neural Contextual Bandit algorithm in Alg. 3, we use an $\epsilon$-greedy strategy to iteratively sample the $D$ configurations: at each of the $D$ iteration, with probability $\epsilon = 0.1$ we select a configuration uniformly at random among the unselected ones in $A$, and with probability $1 - \epsilon$, we select the configuration with the highest reward point estimation among the unselected ones in $A$. The $\epsilon$-greedy strategy does

not require confidence bound estimation and has been commonly used in deep reinforcement learning (such as deep Q Learning [39]) when the action space is large. This ablation allows us to assess the effectiveness of upper confidence bound (UCB) estimation for better a exploration-exploitation trade-off in neural contextual bandit.

### A.7.2 Additional ablation results and analysis

**Ablation: Different configuration subspaces for different configuration update steps**

Our complete method (L2Sep) in the main paper constructs the subspace $A$ only once at the initial update for computational efficiency benefits. We present the ablation where we construct a new subspace $A_2$ at the second update. Specifically, for each MILP instance $x \in \mathcal{K}_{small}$, we use the first trained predictor $\tilde{f}^1_{\theta^T}$ to set the initial configuration at separation round $n_1$ and hold the configuration between separation rounds $[n_1, n_2]$. At the start of separation round $n_2$, the MILP instance is updated to $\tilde{x}$, which combines the original MILP $x$ with the newly added cuts. For ease of explanation, we let $\mathcal{K}_{small,2}$ denote the updated training set of MILP instances where each instance $\tilde{x}$ starts at separation round $n_2$. We run Alg. 1 on $\mathcal{K}_{small,2}$ to select a configuration subspace $A_2$ based on the time improvement when we set the configuration at separation round $n_2$ and maintain it until the solving process terminates. We use the same set of parameters (including filtering threshold $b$, size of the subspace $|A|$, and size of the initial samples $|S|$) as when we construct $A$.

We first compare our choice of reusing the previous subspace $A$ with the ablation method of updating the subspace to $A_2$ by evaluating the two terms in Eq. (3) of the main paper that balances training set performance and generalization: (1) the empirical performance of the instance-aware ERM predictors, $\hat{\Delta}(\tilde{f}^{ERM}_{A,2})$ and $\hat{\Delta}(\tilde{f}^{ERM}_{A_2,2})$, which we denote as $A$: $1st$ and $A_2$: $1st$, and (2) the average empirical instance-agnostic performance, $\frac{1}{|A|}\sum_{s\in A}\hat{\bar{\delta}}(s)$ and $\frac{1}{|A_2|}\sum_{s\in A_2}\hat{\bar{\delta}}(s)$, which we denote as $A$: $2nd$ and $A_2$: $2nd$, on the updated training set $\mathcal{K}_{small,2}$. The ERM predictors $\tilde{f}^{ERM}_{A,2}$ and $\tilde{f}^{ERM}_{A_2,2}$ set the configuration once at separation round $n_2$ by choosing optimally within the subspace $A$ and $A_2$ for each instance in the updated training set $\mathcal{K}_{small,2}$.

Table 12: Comparison of (1) the empirical perf. of the instance-aware ERM predictor $\hat{\Delta}(\tilde{f}^{ERM}_A)$ and $\hat{\Delta}(\tilde{f}^{ERM}_{A,2})$, denoted as $A$: $1st$ and $A_2$: $1st$, and (2) the average empirical instance-agnostic perf. $\frac{1}{|A|}\sum_{s\in A}\hat{\bar{\delta}}(s)$ and $\frac{1}{|A_2|}\sum_{s\in A_2}\hat{\bar{\delta}}(s)$, denoted as $A$: $2nd$ and $A_2$: $2nd$, on the reused subspace $A$ and the updated subspace $A_2$.

|  | $A$: $1st$ | $A$: $2nd$ | $A_2$: $1st$ | $A_2$: $2nd$ |
|---|---|---|---|---|
| Bin. Pack. | 63.1% | 28.9% | 68.9% | 39.3% |
| Pack. | 47.2% | 5.8% | 44.3% | 8.2% |
| Indep. Set | 76.1% | 53.9% | 77.9% | 57.8% |
| Fac. Loc. | 37.7% | 7.3% | 46.7% | 13.1% |

Table 12 displays the relevant statistics on the four ablations MILP classes. We observe an overall decrease in both terms when we re-use the subspace $A$ instead of using the updated subspace $A_2$, although the difference is relatively small. Notably, in the Packing class, the $1^{st}$ term of the reused subspace $A$ is higher than that of the updated subspace $A_2$. This is because the filtering criterion for the updated subspace $A_2$ excludes certain configurations that enhance the instance-aware performance of the ERM predictor, but have low instance-agnostic performance. While these configurations pass the filtering criterion during the initial update, they are subsequently filtered out when we construct the updated space $A_2$ with a slight sacrifice in instance-aware performance.

We further follow our reported method (L2Sep) to train a second configuration predictor $\tilde{f}^{2,2}_{\theta^T}$ using neural UCB (Alg. 3) within the updated subspace $A_2$. We denote the updated method as L2Sep+. The performance results on the four ablation benchmarks are reported in Table 13.

We find that our reported model L2Sep (with a single configuration subspace $A$) exhibits similar performance as the updated model L2Sep+ (with the subspace $A$ for the first update and an updated subspace $A_2$ for the second update), albeit L2Sep+ performing slightly better. This observation validates our decision to reuse the subspace $A$ for the second configuration update, as it offers computational efficiency advantages by reducing the number of MILP solver calls during training (by avoiding a second invocation of Alg 1). We attribute this outcome to (i) the diversity of configurations within the subspace $A$, and (ii) the presence of similar characteristics within a solve for the same MILP instance. Combined, these factors allow the subspace $A$ constructed during the initial update

Table 13: Performance (median, mean, interquartile mean, and standard deviation) of our method where we use the same subspace $A$ for both $k = 2$ updates (L2Sep) v.s. we update a new subspace $A_2$ for the second update (L2Sep+).

| | Bin. Pack. | | | | Pack. | | | |
|---|---|---|---|---|---|---|---|---|
| **Method** | Median | Mean | IQM | STD | Median | Mean | IQM | STD |
| L2Sep | 42.3% | 33.0% | 40.5% | 34.2% | 28.5% | 17.7% | 25.2% | 39.3% |
| L2Sep+ (New A) | 43.1% | 34.1% | 41.1% | 34.9% | 29.7% | 19.6% | 28.1% | 38.5% |
| | Indep. Set | | | | Fac. Loc. | | | |
| | Median | Mean | IQM | STD | Median | Mean | IQM | STD |
| L2Sep | 72.4% | 60.1% | 69.8% | 27.8% | 29.4% | 18.2% | 27.5% | 39.6% |
| L2Sep+ (New A) | 68.7% | 61.8% | 67.7% | 25.1% | 29.8% | 23.0% | 28.7% | 32.1% |

to effectively cover the high-performance configurations across various separation rounds.

We also observe that L2Sep+ with the updated subspace $A_2$ demonstrates more significant improvements in mean performance compared to other metrics. We believe this is also because the instance-agnostic performance of a small number of configurations in the first subspace $A$ may degrade during the second update step, resulting in subpar performance on a few outlier instances that negatively impact the mean (see a similar discussion in Appendix A.8.1). However, by reconstructing the second subspace $A_2$, we effectively eliminate those configurations through the second filtering pass, thereby enhancing the robustness of subspace $A_2$ to outlier instances. Hence, the choice to update the second subspace involves a trade-off between computational efficiency and robustness on the outlier instances, and ultimately, should be decided based on the characteristics of the specific real-world application when deploying our method.

### A.8 Detailed Experiment Results

#### A.8.1 Interquartile mean (IQM) and mean statistics

While we report the median and standard deviation in Table 1 and 3 of the main paper, we provide the mean and interquartile mean (IQM) statistics in the following tables 14, 15 and 16. We observe that the interquartile mean performance closely aligns with the median performance reported in the main paper. Meanwhile, the mean improvements are lower than the interquartile mean for all methods. Upon examining the performance of individual instances, we observe that the performance degradation comes from a small number of outlier instances with negative time improvement; on the majority of instances, our learning method is able to achieve significant improvement from SCIP default. Addressing such outlier instances is left as a future work.

Table 14: **Tang et al.** The IQM and mean of the absolute solve time of SCIP default and relative time improvement of different methods across the test set (higher the better, best are bold-faced).

|  | | Packing | | Bin. Packing | | Max. Cut | |
|---|---|---|---|---|---|---|---|
|  | Method | IQM | Mean | IQM | Mean | IQM | Mean |
|  | Default Time (s) | 9.13s | 17.75s | 0.096s | 0.14s | 1.77s | 1.80s |
| Heuristic Baselines | Default | 0% | 0% | 0% | 0% | 0% | 0% |
|  | Random | -102.5% | -117.9% | -112.2% | -122.4% | -143.8% | -131.4% |
|  | Prune | 6.4% | 1.1% | 16.5% | 17.0% | 4.3% | 12.3% |
| Ours Heuristic Variants | Inst. Agnostic ERM Config. | 17.9% | 13.1% | 34.9% | 33.3% | 70.2% | 68.7% |
|  | Random within Restr. Subspace | 17.6% | 12.2% | 28.2% | 25.0% | 67.2% | 66.6% |
| Ours Learn | L2Sep | **25.2%** | **17.7%** | **40.5%** | **33.0%** | **71.8%** | **68.9%** |

Table 15: **Ecole.** The IQM and mean of the absolute solve time of SCIP default and relative time improvement of different methods across the test set (higher the better, best are bold-faced).

|  | | Indep. Set | | Comb. Auction | | Fac. Location | |
|---|---|---|---|---|---|---|---|
|  | Method | IQM | Mean | IQM | Mean | IQM | Mean |
|  | Default Time (s) | 17.52s | 67.18s | 2.81s | 4.18s | 63.58s | 77.90s |
| Heuristic Baselines | Default | 0% | 0% | 0% | 0% | 0% | 0% |
|  | Random | -101.5s | -111.5% | -132.0% | -129.2% | -125.3% | -128.6% |
|  | Prune | 14.5% | 15.0% | 12.2% | 14.6% | 24.2% | 14.3% |
| Ours Heuristic Variants | Inst. Agnostic ERM Config. | 57.2% | 51.5% | 60.9% | 56.8% | 12.4% | 10.9% |
|  | Random within Restr. Subspace | 50.0% | 30.0% | 59.2% | 56.4% | 17.8% | 13.7% |
| Ours Learn | L2Sep | **69.8%** | **60.1%** | **65.9%** | **62.2%** | **27.5%** | **18.2%** |

Table 16: **Real-world MILPs.** The IQM and mean of the absolute solve time of SCIP default and relative time improvement of different methods across the test set (higher the better, best are bold-faced).

| | Method | NN Verif. | | MIPLIB | | Load Balancing | |
|---|---|---|---|---|---|---|---|
| | | IQM | Mean | IQM | Mean | IQM | Mean |
| | Default Time (s) | 33.48s | 36.99s | 16.32s | 45.50s | 32.47s | 32.92s |
| Heuristic Baselines | Default | 0% | 0% | 0% | **0%** | 0% | 0% |
| | Random | -178.0% | -154.6% | -161.7% | -150.1% | -300.1% | -229.6% |
| | Prune | 30.8% | 24.4% | 5.2% | -30.5% | -14.7% | -71.1% |
| Ours Heuristic Variants | Inst. Agnostic ERM Config. | 30.2% | 25.0% | 3.4% | -19.1% | 11.2% | 12.8% |
| | Random within Restr. Subspace | 29.9% | 26.1% | -11.3% | -31.4% | 10.0% | 8.4% |
| Ours Learn | L2Sep | **34.8%** | **29.9%** | **11.9%** | -8.0% | **21.1%** | **18.6%** |

## A.8.2 Result contextualization

A previous work by Paulus et al. [42] evaluates their learning-based method for cutting plane selection on the NN Verfication dataset. As shown in Table 3 of their paper, their best model achieves a median solve time of 20.89s, whereas the default SCIP solver takes a median solve time of 23.65s, resulting in a median relative speed up of 11.67%. While the comparison is far from perfect, our method achieves a higher median relative time improvement of 37.5%. We note that it is reasonable for their reported absolute solve time to be different from ours due to differences in computational machines. We also perform a rough comparison to the MIPLIB results reported in Wang et al. [54], which, same as Paulus et al. [42], learns to select cutting planes. In Table 1 of their paper, they report SCIP default takes $256.58s$ and $164.61s$ on average to solve two small homogenous MIPLIB subsets, whereas their cutting plane selection method improves the solve time to $248.66s$ and $162.96s$, leading to a $3\%$ and $1\%$ improvement. Although not a perfect comparison, our L2Sep achieves a higher median time improvement of 12.9% (and an interquartile mean time improvement of 11.9%) on our larger heterogeneous subset (See Appendix A.6.5 for a detailed dataset description).

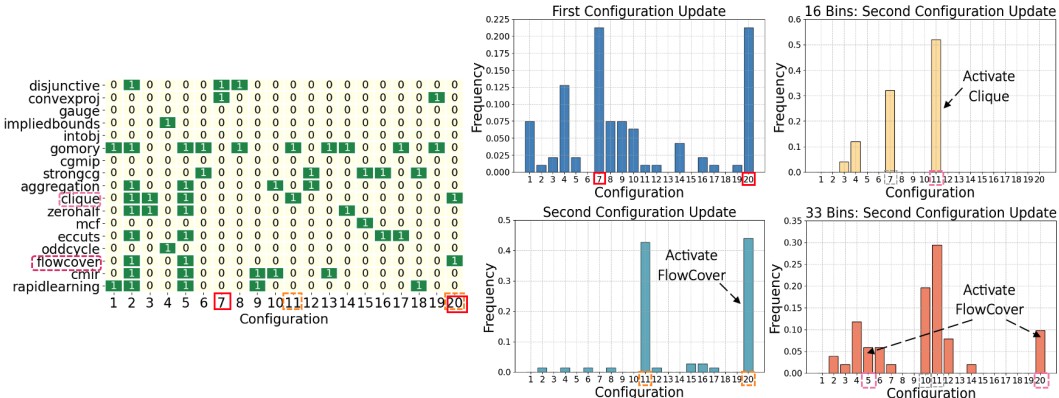

Figure 8: **Bin Packing Interpretation.** (Left / Heatmap) Each row is a separator, and each column is a configuration in our restricted space. Green and yellow cells indicate activated and deactivated separators by each configuration, respectively. (Middle / Histogram Left Column) The frequency of each configuration selected by our learned model at the $1^{st}$ and $2^{nd}$ config. update on the original test set with *66 bins*. High frequency configurations are marked with red and orange squares, respectively. The config. indices in the heatmap and the histograms align. (Right / Histogram Right Column) The frequency of each configuration selected at the $2^{nd}$ config. update (the $1^{st}$ update has similar results) when we gradually decrease the number of bins (bottom: *33 bins*; top *16 bins*). We observe the prevalence of FlowCover and Clique decreased and increased, respectively.

### A.8.3    Interpretation analysis: L2Sep recovers effective separators from literature

In Fig. 8, 9, 10, and 11, we provide visualizations of (1) the restricted configuration subspace $A$, as shown in the heatmap plots, and (2) the frequency for each configuration to be selected by L2Sep on the test set, as shown in the histogram plots, for all MILP classes that we study. As described in the main paper, for Bin Packing, Independent Set, and MIPLIB, the visualization provides meaningful interpretations that recover known facts from the mathematical programming literature.

Besides the known results, we also observe some intriguing unexpected scenarios from the visualizations. For Independent Set, L2Sep deactivates all separators with a frequency of 20% at the $2^{nd}$ configuration update, whereas all selected configurations at the $1^{st}$ update activate a substantial amount of separators. It is an interesting question to investigate why it is better to deactivate all separators for a certain subset of Independent Set instances at later separation rounds.

For Maximum Cut, OddCycle [10, 29] and ZeroHalf [11] are known to be effective in the literature. Interestingly, none of the selected configurations activate ZeroHalf for both configuration updates; OddCycle is also completely deactivated for the $1^{st}$ update, but is activated with a frequency of 14% at the $2^{nd}$ update. Meanwhile, we observe that Disjunctive, FlowCover, and Aggregation separators are more frequently selected.

We hope that by providing the visualization results, L2Sep can serve as a driver of future works on improved (theoretical) polyhedral understanding of different MILP classes, and potentially seed investigations (empirical and theoretical) for nonstandard, newly-proposed problems (e.g. NN Verification) where few analyses exists.

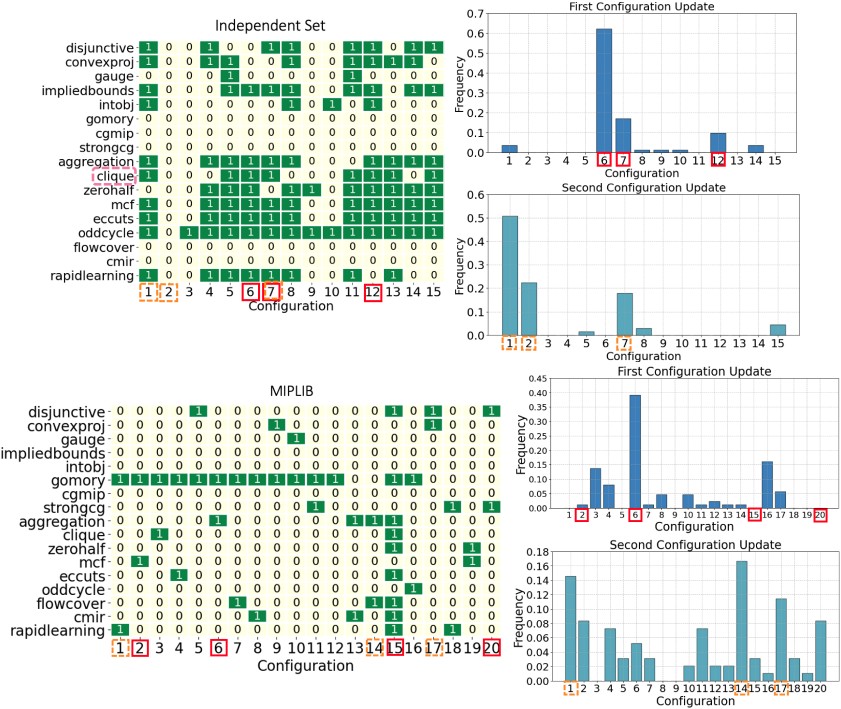

Figure 9: **(Top) Independent Set. (Bottom) MIPLIB.** The same set of figures as Fig. 8 (Left / Heatmap) and (Middle / Histogram Left Column). See interpretations in the main paper.

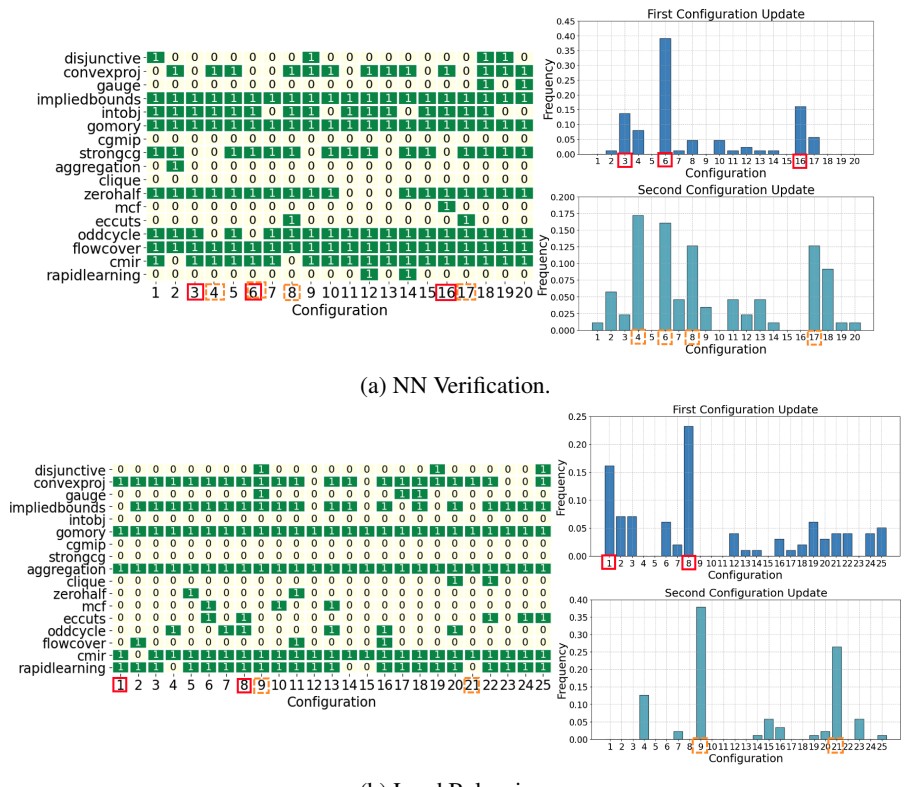

(a) NN Verification.

(b) Load Balancing.

Figure 10: **Other real-world MILP classes.** The same set of figures as Fig. 8 (Left / Heatmap) and (Middle / Histogram Left Column).

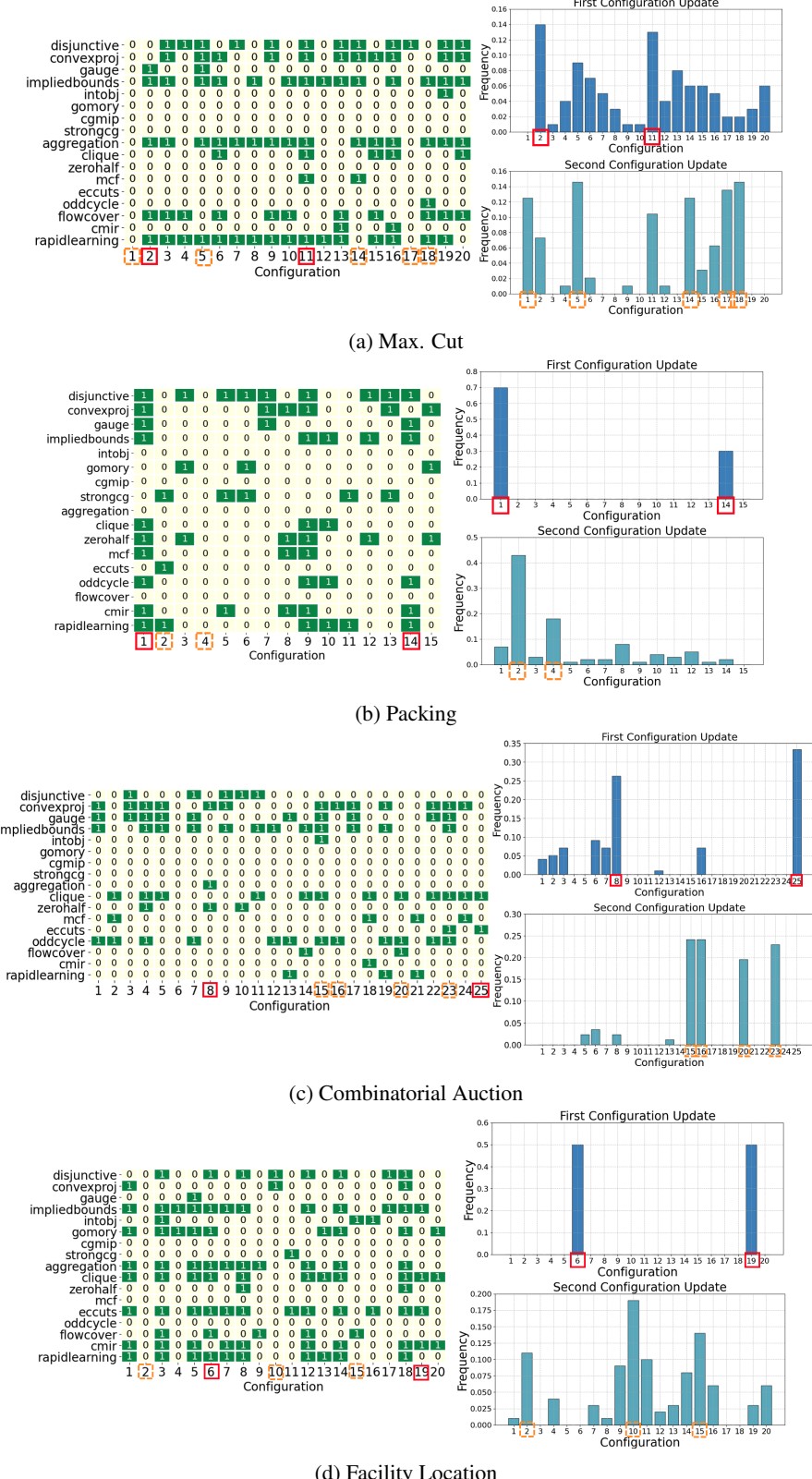

(a) Max. Cut

(b) Packing

(c) Combinatorial Auction

(d) Facility Location

Figure 11: **Other Tang et al. and Ecole MILP classes.** The same set of figures as Fig. 8 (Left / Heatmap) and (Middle / Histogram Left Column).

Table 17: **Alternative objective (relative gap improvement).** Absolute gap of SCIP default under the fixed time limit, and the mean (lower the better, best are bold-faced) and standard deviation of the relative gap improvement of different methods with respect to SCIP default.

| | Methods | Pack. | Comb. Auc. | Indep. Set | NNV | Load Balancing |
|---|---|---|---|---|---|---|
| | Time Limit (s) | 4.4s | 1.4s | 8.2s | 16s | 16s |
| | Default Gap | 9.1e-4 (9.3e-4) | 0.060 (0.098) | 0.057 (0.059) | 0.50 (0.80) | 0.32 (0.13) |
| Heuristic Baselines | SCIP Default | 0% | 0% | 0% | 0% | 0% |
| | Random | -37.1% (41.7%) | -27.3% (69.1%) | -23.2% (44.1%) | -40.3% (72.8%) | -48.0% (35.8%) |
| Ours Heuristic Variants | Inst. Agnostic Configuration | 11.9% (38.4%) | 52.4% (45.3%) | 23.5% (34.5%) | 33.6% (72.1%) | 14.0% (18.9%) |
| | Random within Rest. Subspace | 10.1% (42.5%) | 54.1% (45.1%) | 21.6% (33.7%) | 24.8% (75.9%) | 9.5% (17.6%) |
| Ours Learned | **L2Sep** | **15.4% (40.0%)** | **68.8% (38.2%)** | **29.6% (34.7%)** | **36.0% (68.2%)** | **34.2% (27.5%)** |

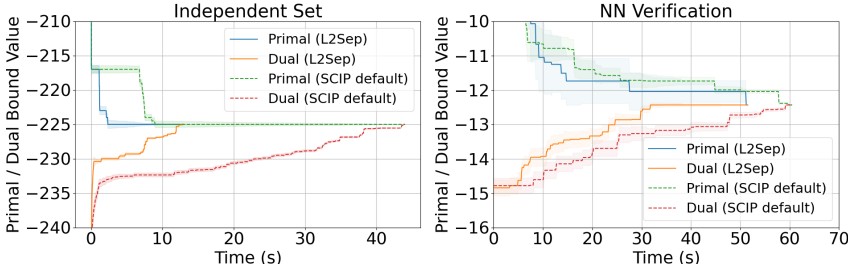

Figure 12: Primal-dual bound curves (median and standard error) for L2Sep and SCIP default on Independent Set and NN Verification. L2Sep can effectively tighten the dual bound faster.

### A.8.4 The immediate and multi-step effect of separator configuration in the B&C process

Intelligently configuring separators have both immediate and multi-step effects in the B&C process:

**Immediate:** Some separators take a long time to run, but generate mostly low-quality cuts that are ultimately never selected by the downstream cut selector. Deactivating those separators leads to an immediate time improvement by reducing the time to generate the cut pool.

**Multi-step:** Improved separator configuration can tighten the dual bound faster through better-selected cuts; it may also accelerate other B&C components such as branching (e.g. strong branching requires solving many children LPs and hence may benefit from tighter dual bounds).

The Table 18 presents the total solve time and total separator execution time for our complete method L2Sep and SCIP default on several MILP classes. We report the median and standard deviation evaluated on 100 instances for each class. L2Sep significantly reduces the total separator execution time. Upon closer examination, we find that L2Sep adeptly deactivates expensive yet ineffective separators while activating effective ones.

In Fig. 12, we plot the primal-dual bound curves (median and standard error) of L2Sep and SCIP default on Independent Set and NN Verification. The significantly faster dual bound convergence of L2Sep demonstrates the multi-step effect of improved separator configurations. We further summarize the synergistic interaction effects between separator selection and other B&C components (branching, dual LP) in Table 19. Notably, even though our method does not modify branching, the branching solve time is reduced.

Table 18: **Immediate Effect of Separator Configuration.** The median (lower the better) and standard deviation (in parentheses) of the total solve time and total separator execution time for L2Sep and SCIP default.

|  | Total Solve Time (L2Sep) | Total Separator Execution Time (L2Sep) | Total Solve Time (SCIP Default) | Total Separator Execution Time (SCIP Default) |
|---|---|---|---|---|
| Comb. Auc. | **0.65s** **(2.36s)** | **0.02s** **(0.054s)** | 3.01s (4.60s) | 1.39s (1.14s) |
| Indep. Set | **3.81s** **(116.42s)** | **0.35s** **(9.94s)** | 13.16s (120.89s) | 7.38s (37.94) |
| NNV | **20.75s** **(18.56s)** | **0.16s** **(0.13s)** | 34.76s (25.05s) | 6.58s (8.05s) |

Table 19: **Multi-Step Effect of Separator Configuration.** The median (lower the better) and standard deviation (in parentheses) of strong branching time, pseudocost branching time, and dual LP time for L2Sep and SCIP Default.

|  | Comb. Auc. | | Indep. Set | | NNV | |
|---|---|---|---|---|---|---|
|  | L2Sep | SCIP Default | L2Sep | SCIP Default | L2Sep | SCIP Default |
| **Strong Branching Time** | **0.31s** **(1.15s)** | 0.41s (1.79s) | **2.82s** **(21.44s)** | 3.92s (19.68s) | **6.56s** **(4.06s)** | 8.31s (5.55s) |
| **Pseudocost Branching Time** | **0.35s** **(1.46s)** | 0.45s (2.17s) | **3.01s** **(22.29s)** | 4.6s (21.67s) | **8.18s** **(4.81s)** | 9.69s (6.13s) |
| **Dual LP Time** | **0.08s** **(0.58s)** | 0.18s (0.83s) | **0.3s** **(73.95s)** | 1.14s (55.51s) | **3.67s** **(6.52s)** | 5.07s (5.81s) |

### A.8.5 Alternative objective: relative gap improvement

We analyzed an alternative objective of the relative gap improvement under a fixed time limit. Let $g_0(x)$ and $g_\pi(x)$ be the primal-dual gaps of instance $x$ using the SCIP default and another configuration strategy $\pi(x)$ under a fixed time limit $T$. We define the relative gap improvement as $\delta_g(\pi(x), x) := (g_0(x) - g_\pi(x))/(\max\{g_0(x), g_\pi(x)\} + \epsilon)$. We choose the denominator to avoid division by zero when the instance is solved to optimality.

As seen from Table 17, L2Sep achieves a 15%-68% relative gap improvement over SCIP default. Specifically, the table presents the relative gap improvement (mean and standard deviation) of each method over SCIP default, along with the fixed time limit for various MILP classes (mostly around 50% of medium SCIP default solve time), and the absolute gap of SCIP default at the time limit. In Fig. 13, we further plot histograms of the gap distribution on the entire dataset for L2Sep and SCIP default, where we observe that L2Sep effectively shifts the entire gap distribution to a lower range. These results demonstrate the effectiveness of our method across different objectives, and its ability to improve primal-dual gaps for instances that cannot be solved to optimality within the time limit.

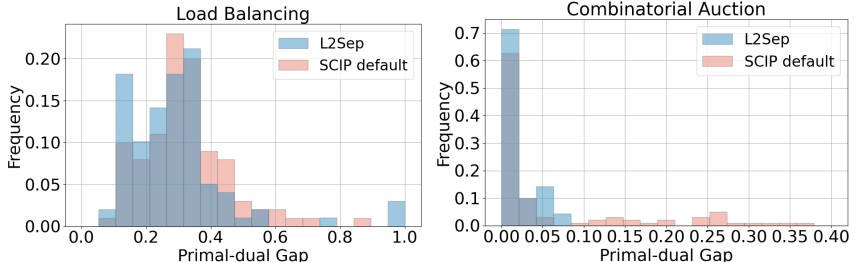

Figure 13: Distributions of absolute primal-dual gaps of L2Sep and SCIP default on Load Balancing and Combinatorial Auction. L2Sep is effective under an alternative objective (relative gap improvement).

## A.9 Limitation.

While SCIP default does not require training as it sets pre-defined priorities and frequencies of separators (the same for all MILP instances), our learning method requires collecting data and fitting models for each MILP class, which introduces a time overhead. However, such a limitation is inherent in all learning for MILP methods [51, 42, 54, 18], as learning methods rely on training models to generalize to unseen test instances. Following these previous works, we train separate models for different MILP classes, although our L2Sep method exhibits a significant time improvement on the heterogeneous MIPLIB dataset, which suggests the possibility of learning an aggregated model for multiple MILP classes to potentially reduce training time. We leave this as a future work.

In the ablation Sec. 6.2 of the main paper, we observe that learning $k = 3$ configuration updates offers limited improvement from our L2Sep method with $k = 2$ updates. While it is beneficial to achieve a significant time improvement with a small number of updates as it simplifies the learning task and reduces training time, the optimal policy for finer-grained control should theoretically yield better performance (smaller approximation error to the optimal policy that allows updates at all separation rounds). Potential future research would involve exploring the learning of more frequent configuration updates, possibly by considering advanced reinforcement or imitation learning algorithms.

Another potential limitation of our learning method (as well as all baseline methods mentioned in the main paper) is that the mean performance tends to be lower than the interquartile mean or median due to the long solve time on a small set of outlier instances which skew the mean. A potential future research direction would involve developing techniques to identify these outlier instances, on which we use SCIP default instead of configuring with learning or heuristics methods.

## A.10 Negative Social Impact.

Application of deep learning in discrete optimization may contribute to increased use of computation for training the models, which would have energy consumption and carbon emissions implications. The characterization and mitigation of these impacts remain an important area of study.