# OpenReview forum: "Learning to Configure Separators in Branch-and-Cut"
_NeurIPS.cc/2023/Conference — NeurIPS 2023 poster_

### Official Review · Reviewer_wvgK · 2023-06-23

**Soundness:** 4 excellent
**Presentation:** 3 good
**Contribution:** 2 fair
**Rating:** 4
**Confidence:** 4

**Summary:**

The authors use machine learning to decide how and when to toggle the on/off switches for different cutting plane families provided by SCIP, the fastest open-source MIP solver. Their learning algorithm outperforms the default setting of SCIP on various benchmark sets.

**Strengths:**

The techniques used to deal with the high dimensional combinatorial space of possible configurations are interesting. The results are promising and yield speedups over SCIP default on established benchmark sets.

The interactions between different cut families is not an extremely well understood topic. It is nice to know that machine learning can help decide what families to activate, and when.


**Weaknesses:**

Ultimately I don’t think the proposed methodology is too novel, since it boils down to learning how to set on-off toggles for a variety of cut families. I acknowledge that this toggling is a critical component of tuning MIP solvers, but since this paper doesn’t mention what cut families ended up being selected and most useful for the different problem instances, I believe it leaves the most interesting question of what separator families worked best for which problems on the table. So methodologically this paper doesn’t seem too different from an array of previous algorithm configuration papers for MIP (e.g. Hydra-MIP by Xu, Hutter, Hoos, Leyton-Brown) that aim to tune parameters of a MIP solver from past data, but without any deeper principled investigation of what the parameters are actually doing. Hypothetically, it could be the case that turning off SCIP’s Gomory cut generation helps out for a class of MIPS, but maybe if the actual Gomory cut generation was tweaked performance would improve. Learning to toggle does not yield any such insights, as far as I can tell.

**Questions:**

The title/phrase “learning to separate” slightly misleading. The *separation problem* refers to the specific problem of generating a cut (from some class of cuts) that cuts off the LP optimum. The authors here are not concerned with that problem, rather they are concerned with the on/off toggle for a particular family of cuts that is being generated via SCIP’s separation routines.

The authors discuss generalization guarantees but do not cite or compare to any prior work on such theoretical guarantees for cutting planes/integer programming/tree search. It might be worth mentioning how this compares to that work and where it is different (ostensibly “learning to toggle” falls into some of the frameworks studied previously).

As mentioned previously, one of the most interesting questions that is completely missing: what cut families ended up being selected and working well for the different problem instances? To me this would be the most interesting set of conclusions, since it is well known that MIP solvers need to be tuned to yield improved performance. The question of what cuts work best for what problems is much more elusive. Overall, I would be much more in favor of acceptance if there was some discussion about this aspect. Presumably this question is already answered by the experiments the authors ran, and including some observations here would be great.

Given that the authors are controlling granular on/off parameters (and not fundamentally modifying the underlying algorithms), why not compare to a state-of-the-art solver like Gurobi or CPLEX, which are free for academic use and significantly faster than SCIP?

---

> ### Author Rebuttal · Authors · 2023-08-09
>
> We thank the reviewer for the actionable feedback in terms of interpreting the results, using other MILP solvers, and providing pointers to relevant theoretical works. We have made great efforts to address the reviewer’s concerns in our response below and have included additional experiments in the general response, covering (1) interpretations of effective separators for each MILP class from our learning results (2) applying our method to a different MILP solver Gurobi, and (3) applying our method to a different metric (relative gap improvement under a fixed time). We hope that the Reviewer will take these new results into account and increase our score if we have adequately addressed the main concerns.
>
> > As mentioned previously, one of the most interesting questions that is completely missing: what cut families ended up being selected and working well for the different problem instances?
>
> We sincerely appreciate the reviewer’s excellent suggestions on interpreting our learning results. We provide visualizations and interpretations in General Response [GR1]. These visualizations lead to several intriguing observations, including (1) our learned model can automatically select separator families that are known to be effective for certain MILP class, and (2) our learned model can differentiate heterogeneous instance class and select customized separators tailored to different types of MILP instances.
>
> We plan to include visualizations for all MILP classes in the Appendix of the updated paper. Given the alignment of effective separator families on standard MILP benchmarks between our learning model and the existing literature, we believe our learned model can serve as a valuable tool for guiding the automatic selection of separator families for different MILP classes, and inspiring the mathematical programming community to further investigate the interconnection between certain separators and MILP instances as suggested by our learning method.
>
> > Given that the authors are controlling granular on/off parameters (and not fundamentally modifying the underlying algorithms), why not compare to a state-of-the-art solver like Gurobi or CPLEX, which are free for academic use and significantly faster than SCIP?
>
> SCIP allows us to update separator configurations multiple times during a solve process (by modifying the source code), while state-of-the-art solvers Gurobi does not (separator configurations have to be fixed before the solve starts). Nonetheless, we can apply our method to Gurobi by configuring separators once before the solve starts. Inspired by the reviewer’s question, we perform this additional experiment in General Response [GR2]. We are delighted to report that our method is also effective in accelerating Gurobi.
>
> > The authors discuss generalization guarantees but do not cite or compare to any prior work on such theoretical guarantees for cutting planes/integer programming/tree search. It might be worth mentioning how this compares to that work and where it is different.
>
> We thank the reviewer for providing references to relevant papers. We will include them in the updated paper.
>
> The most relevant work [1] studies generalization of portfolio-based algorithm selection, where the procedure first selects a subset of algorithm parameter settings, and then, for a given problem instance, uses an algorithm selector to choose a parameter setting from the portfolio. We summarize the key differences below:
>
> - Our theoretical generalization analysis **directly informs** our empirical configuration subspace restriction algorithm, where we design a filtering criterion to improve the generalization bound. In contrast, the previous theoretical work analyzes the generalization bound on a *given* subspace construction procedure; however, their bound is not informative for designing the construction procedure, as it contains an abstract constant (representing the number of piecewise constant regions in the performance function [2]) which is unknown in practice and hence cannot be applied empirically.
> - the generalization bound in the previous work does not consider the influence of the quality and diversity of the parameter settings within a subset of a fixed size, which our generalization bound captures. In our experiments, we demonstrate that the quality of each configuration is important for a fixed size configuration subspace (see our ablation “Greedy Restr.” in Table 2 which does not apply our filtering criterion to consider individual configuration’s quality, and leads to suboptimal results).
>
> Additional theoretical works investigate the generalization bound of the branch-and-cut framework, mainly focusing on cut selection [3, 4]. However, the analyses either rely on the structure of a specific MILP cut family (e.g. Chvátal-Gomory) or study how a selected cut affects the LP relaxation’s optimal solution, which cannot be generalized to the upstream, higher-level task of separator configuration.
>
> *[1] Balcan, Maria-Florina, Tuomas Sandholm, and Ellen Vitercik. "Generalization in portfolio-based algorithm selection." Proceedings of the AAAI Conference on Artificial Intelligence (2021).*
>
> *[2] Balcan, Maria-Florina, et al. "How much data is sufficient to learn high-performing algorithms? Generalization guarantees for data-driven algorithm design." Proceedings of the 53rd Annual ACM SIGACT Symposium on Theory of Computing. (2021).*
>
> *[3] Balcan, Maria-Florina F., et al. "Sample complexity of tree search configuration: Cutting planes and beyond." Advances in Neural Information Processing Systems 34 (2021).*
>
> *[4] Balcan, Maria-Florina F., et al. "Structural analysis of branch-and-cut and the learnability of gomory mixed integer cuts." Advances in Neural Information Processing Systems 35 (2022).*
>
> > The title/phrase “learning to separate” is slightly misleading.
>
> We are happy to change the title to “Learning to configure separators in branch-and-cut”.

---

> > ### Comment · Reviewer_wvgK · 2023-08-10
> >
> > I greatly appreciate the authors' amazingly thorough response. The qualitative results are very interesting, and it is especially fascinating to me that the L2Sep method "discovers" classical knowledge.
> >
> > Coupled with this, I find the message that L2Sep meaningfully shows when and where to toggle specific cut families to be significant and interesting, given that the "when to cut" question is equally not-well-understood as the "how to cut" question, and has received significantly less study. My initial score of 4 was probably too harsh, and I would be happy to raise my score to a 6/7.
> >
> > I do think the title change proposed by the authors would be appropriate.

---

> > > ### Author Response · Authors · 2023-08-10
> > > **Thank you for your feedback and suggestions!**
> > >
> > > We are thankful to the reviewer for taking a careful look at our rebuttal and taking the time to revise their assessment. We're happy that our rebuttal alleviated the reviewer's main concerns, and we will make the title change in the updated paper. Thank you again for your detailed feedback and suggestions throughout the whole process!

---

### Official Review · Reviewer_TfAQ · 2023-07-05

**Soundness:** 3 good
**Presentation:** 4 excellent
**Contribution:** 4 excellent
**Rating:** 8
**Confidence:** 4

**Summary:**

The paper proposes a pipeline for a learning-enhanced cut separation management in modern MIP solvers (specifically, in the academic solver SCIP). To pick a promising subset of the large number of possible settings, the authors derive a data-driven and theoretically motivated method to focus on specific configurations that are expected to perform well. The actual policy on how cut separators are used during a MIP solving run are then learned over the restricted configuration space. The experimental results show significant solve-time improvements for a variety of benchmark MIP classes and test sets; this holds both when learning within MIP subclasses and when learning within heterogeneous benchmark sets such as MIPLIB. The presented work has the potential to replace current fixed, heuristic solver settings for cutting plane separation by carefully learned policies that often yield large speed-ups, which in turn can increase the size of problem instances that can be solved to optimality, and therefore represents a very notable advancement in data-driven MIP solving techniques.

-- update: I have read and acknowledged all other reviews and the authors rebuttals, see discussion. --

**Strengths:**

Besides the impressive empirical results, the theoretical justification/motiviation for the configuration pre-selection is a strength of this work. Moreover, although the work spans quite many different and subtle aspects of the intricate workings of MIP solvers and employs different machine learning techniques, the authors did an excellent job in condensing their work into an understandable and well-written main paper; the supplementary material, while containing the technical proofs, is largely additional information that well complements the main paper but is mostly not necessary to follow the main paper.

**Weaknesses:**

It appears to have become common practice to submit papers to NeurIPS (and ICML) whose actual, main content is put in a separate "Supplementary Material" document whose length far exceeds that of the supposed main paper. This paper is only a partial exception -- the main paper provides enough information and details to stand on its own except for the most technical bits, which can then be found in the very long Appendix (supplementary document; three times as long as the main paper) along with a host of additional information. Thus, it may be considered a weakness of the paper to have such a long Appendix, because this format bears the danger of the formally most important parts of the work (proofs; algorithm details and specific setups) not being reviewed thoroughly due to the short review period and high review load of reviewers at these conferences. I cannot exclude myself from this -- I simply did not have the time to rigorously check all the details in the long supplementary document, and therefore cannot give a definitive answer regarding the proofs' correctness beyond "believing" everything appears to be well in order. In this regard, I cannot help but wonder if a full journal paper would not be the better way to publish results that simply do not fit into the 9-page limit. But, again, I have seen papers that exploit the main paper/supplementary material split in a much worse way; for the present work, it actually hardly bothered me because the main paper is nearly self-sufficient. Nevertheless, if I have to point to a weakness, this aspect is what stands out to me.

**Questions:**

- there are several (minor) typos throughout paper and supplementary document (e.g., no comma after "e.g.", "c.f." or "v.s." instead of "cf." and "vs.", "constraint" <-> "constrained" (caption of Fig.2), "contrary" <-> "contrast" (l.193), "included A" <-> "included in A" (l. 239), "which we provide comparision" <-> "for which we provide comparison" (l. 311), "U Stuhl" (ref. [47] -- what's the first name?))
- some wordings are a bit inaccurate (at least when not having seen the supplementary document yet, i.e., when reading just the first paper):
  * l.19: "lower bound" -- this implies that the problem is a minimization problem, which is not specified here (only in the supplement). Maybe clarify, or use "dual bound"?
  * l. 66 and again later: The product-sign (\prod , large Pi) should be a *Cartesian* product-sign
  * Prop. 1 is referred to as "lemma" in the paragraph preceding it.
   * Sect. 3.2: "network" kind of drops out of nowhere -- please clarify a bit here that neural networks are used for the discussed task
   * in Tab. 1 and 3 at least: "higher the better" only applies to the median; for standard deviation, lower is better!
   * is the reported standard deviation given in percent deviation from the default or as percent variation from the median values? how often, if at all, does reconfiguration actually slow down the solver? what are the mean/average deviations from default?
- it would be good to clarify in the main paper that the separation rounds after which configuration updates occur are specified in advance; this only becomes clear in the supplementary document, so reading the main paper, one wonders how this is decided and/or if it is part of the learning procedure to decide when to update configurations. Also, are there cases in which the MIP solver terminates before the number k of updates has been performed?
- have you tried reverting to the default configuration as the last update?
- which LP solver was used by SCIP to solve the relaxations? SoPlex?
- were all MIPs solved in single-thread mode?

**Limitations:**

Limitations have been appropriately addressed (at least in the supplementary material).

---

> ### Author Rebuttal · Authors · 2023-08-09
>
> We are delighted and truly appreciate the reviewer‘s positive feedback on our work. We made a great effort to integrate empirical methodology with theoretical justification, aiming at bridging an important gap in the existing learning for MILP literature that has been predominantly empirical. We provide a response to the reviewer’s questions below, and we include additional experiments as suggested by other reviewers in the general response, covering (1) interpretations of effective separators for each MILP class from our learning results (2) applying our method to a different MILP solver Gurobi, and (3) applying our method to a different metric (relative gap improvement under a fixed time).
>
> > is the reported standard deviation given in percent deviation from the default or as percent variation from the median values? how often, if at all, does reconfiguration actually slow down the solver? what are the mean/average deviations from default?
>
> The following table shows the percentage of test instances in each MILP class for which our learned configuration improves over SCIP default (% win). We observe that, while not always, our learned model does accelerate SCIP default for the majority of instances.
>
> ||Bin. Pack.|Max. Cut|Pack.|Comb. Auc.|Indep. Set|Fac. Loc.|NNV|MIPLIB|Load Balancing|
> |-|-|-|-|-|-|-|-|-|-|
> |% win (ours > default)|91.2%|100%|75%|98.9%|95.7%|74%|88.5%|67.5%|84%|
> |standard deviation (from mean), Table 1 |(34.2%)|(11.3%)|(39.3%)|(26.2%)|(27.8%)|(39.6%)|(33.9%)|(73.1%)|(20.3%)|
> | deviation from median|(39.0%)|(11.6%)|(42.8%)|(29.3%)|(37.0%)|(43.5%)|(35.5%)|(78.8%)|(22.1%)|
> | deviation from default|(52.1%)|(68.4%)|(44.2%)|(67.2%)|(66.6%)|(46.3%)|(45.2%)|(75.3%)|(32.2%)|
> ||
>
> In our paper, we report the standard deviation from the mean $\sqrt{\frac{1}{N}\sum\limits_{i=1}^{N} (\delta_i - \bar{\delta})^2}$ of each method, where $\bar{\delta} = \frac{1}{N}\sum\limits_{i=1}^{N}\delta_i$, and $\delta_i$'s are the relative time improvement of each method from SCIP default on $N$ MILP test instances. In the table above, we further include the deviation from the median $\sqrt{\frac{1}{N}\sum (x_i - x_{median})^2}$ and the deviation from SCIP default $\sqrt{\frac{1}{N}\sum (x_i-0)^2}$ (since the time improvement of default from default is 0%), both for our complete method. We observe that the deviation from the median is similar to the deviation from the mean reported in Table 1, while the larger deviation from SCIP default demonstrates the ability of our method to achieve time improvement from the default parameters.
>
> > it would be good to clarify in the main paper that the separation rounds after which configuration updates occur are specified in advance; this only becomes clear in the supplementary document, so reading the main paper, one wonders how this is decided and/or if it is part of the learning procedure to decide when to update configurations. Also, are there cases in which the MIP solver terminates before the number k of updates has been performed?
>
> We appreciate the reviewer for carefully reviewing the supplementary document. We will add clarifications in the main paper to state that the separation rounds are specified in advance. In our experiments, we set the second separator configuration ($k=2$) to be at around 15%-25% of the average total separation rounds for each MILP classes, and set the third separator configuration ($k=3$ in our Ablation Table 2) to be at around 25%-50%, as the early solving stage (e.g. earlier in the B&B tree) may benefit more from additional configuration updates.
>
> It is an interesting suggestion to consider learning to decide when to update the configurations, but it comes with learning challenges as the search space becomes a larger, joint space of (which separation round to configure $\times$ which configuration to choose). We consider this as a future work to explore additional search space reduction techniques to enable joint learning. While there are instances that terminate before the $k^{th}$ of updates has been performed, they only constitute a very small portion of all instances in our experience. Moreover, such instances that terminate before the $k^{th}$ configuration update likely have a short solve time (due to fewer separation rounds to solve the instance). The model at the $k^{th}$ configuration update can then focus on the harder instances that take longer to solve and further improve their solve time from default.
>
> > have you tried reverting to the default configuration as the last update?
>
> We conduct an experiment where we revert to the default SCIP configuration at two intermediate separation rounds (20 and 40) for Ecole instances (Combinatorial Auction, Independent Set, and Capacitated Facility Location). We compare the time improvement for these two scenarios with our complete method, where we apply learned configuration updates for the entire solve process. Our results (median and standard deviation) are as follows:
>
> ||Comb. Auc.|Indep. Set|Fac. Loc.|
> |-|-|-|-|
> |Revert to default at separation round 20|61.9% (31.3%)|39.3% (53.2%)|21.4% (40.8%)|
> |Revert to default at separation round 40|62.8% (31.4%)|61.3% (90.8%)|27.4% (40.9%)|
> |Ours: L2Sep|**66.2% (39.3%)**|**72.4% (27.8%)**|**29.4% (39.6%)**|
> ||
>
> We observe that the relative time improvement increases as our learned configurations cover more separation rounds,  demonstrating our learned method is able to achieve time improvement throughout the solve process.
>
>
>
> > which LP solver was used by SCIP to solve the relaxations? SoPlex?
> >
> >
> > were all MIPs solved in single-thread mode?
>
> We use the default LP solver by SCIP, which is SoPlex [1], and solve all MIPs in single-thread mode.
>
> *[1] Gamrath, Gerald, et al. "The SCIP optimization suite 7.0." (2020).*
>
>
> > typos and wording
>
> We are very thankful that the reviewer points out typos and suggests wording changes. We will correct them in the updated paper.

---

> > ### Comment · Reviewer_TfAQ · 2023-08-10
> >
> > I thank the authors for their detailed responses to all comments by the other reviewers and myself. Especially the additional insights summarized in the general response will be a valuable addition to the paper.
> >
> > A quick (?) follow-up question regarding the interpretability analysis in GR1: Besides identifying known results/what was known to work well for specific instance classes, did you observe something "unexpected" or "unexplained"?  For example, did some family of cuts consistently appear in selected configurations for an instance class for which it is not known that or why these cuts would be useful? If so, this may warrant a closer (theoretical) inspection of the problem-cut pair in future work, so possibly, L2Sep could also serve as a driver of improved polyhedral understanding fof some problems.
> >
> > Overall, I am strongly in favor of accepting this paper, and will increase my score. I very much hope that the other reviewers see the merits of this work and raise their scores at least into acceptance territory as well.

---

> > > ### Author Response · Authors · 2023-08-10
> > > **Thank you for your feedback!**
> > >
> > > We are very thankful to the reviewer for recommending acceptance of our work! Regarding the reviewer’s followup question, we do observe some intriguing unexpected scenarios from the visualizations.
> > >
> > > For Independent Set (see Rebuttal PDF Fig. 2), L2Sep deactivates all separators with a frequency of 20% at the 2nd config. update, whereas all selected configurations at the 1st update activate a substantial amount of separators. It is an interesting question to investigate why it is better to *deactivate all* separators for a certain subset of Independent Set instances at *later* separation rounds.
> > >
> > > For Maximum Cut, OddCycle [1, 2] and ZeroHalf [3] are known to be effective in the literature. Interestingly, none of the selected configurations activate ZeroHalf for both config. updates; OddCycle is also completely deactivated for the 1st update, but is activated with a frequency of 14% at the 2nd update. Meanwhile, we observe that Disjunctive, FlowCover, and Aggregation separators are more frequently selected. We will provide the visualization in the Appendix.
> > >
> > > In summary, we agree with the reviewer that L2Sep could also serve as a driver of improved (theoretical) polyhedral understanding of some problems. We further believe that L2Sep can be helpful to seed investigations (empirical and theoretical) for nonstandard, newly-proposed problems (e.g. NN Verification) where few analyses exists. We thank the reviewer again for the time taken throughout the process to thoroughly review our work!
> > >
> > > *[1] Boros, Endre, Yves Crama, and Peter L. Hammer. "Chvátal cuts and odd cycle inequalities in quadratic 0–1 optimization." SIAM Journal on Discrete Mathematics 5.2 (1992): 163-177.*
> > >
> > > *[2] Jünger, Michael, and Sven Mallach. "Exact facetial odd-cycle separation for maximum cut and binary quadratic optimization." INFORMS Journal on Computing 33.4 (2021): 1419-1430.*
> > >
> > > *[3] Caprara, Alberto, and Matteo Fischetti. "{0, 1/2}-Chvátal-Gomory cuts." Mathematical Programming 74 (1996): 221-235.*

---

> ### Comment · Reviewer_TfAQ · 2023-08-10
> **Raise score to 8**
>
> I don't seem to be able to edit my review directly; I would raise my rating from 7 to 8.

---

### Official Review · Reviewer_JiCg · 2023-07-07

**Soundness:** 3 good
**Presentation:** 3 good
**Contribution:** 2 fair
**Rating:** 6
**Confidence:** 4

**Summary:**

This paper studies learning to manage separators to improve MILP solvers. Specifically, it learns a policy to determine which separators to use to generate cutting planes. The task is well formulated and experiments on many datasets demonstrate the effectiveness of the proposed model.

**Strengths:**

1.	The paper is well written. Especially the problem is well formulated.
2.	The paper identifies the opportunity of managing separators to improve MILP solvers.
3.	Experiments on many datasets demonstrate the effectiveness.

**Weaknesses:**

In general, I think this paper is an OK work, but it is just borderline to the NeurIPS bar, so I only give borderline accept.

There have been many works that study different parts of branch-and-bound, e.g., cut selection and node selection. Replacing one of the solving stages to a learning method, which leads to improvements compared with heuristic rules, is unsurprising. However, in real applications, it would be impossible to replace every part to learned models because of the memory limitation. Whether the studied part is critical enough in branch-and-bound is an important question. Therefore, separator configuration may not be in a trend of the research community, and thus the impact of the proposed method may be limited.

I would give some suggestions in improving the work. First, the authors can compare the focused task, i.e., separator configuration, with other solving stages to demonstrate its significance. Or the authors can compare the resource usage to show the proposed method is lightweight enough so that it can be used jointly with other methods. Or the authors can try to merge the proposed framework with existing work and conduct enough ablation study. Summarily, the authors should denonstrate the real usefulness of the proposed method beyond only its performance.

**Questions:**

See Weakness.

---

> ### Author Rebuttal · Authors · 2023-08-09
>
> We appreciate the insightful suggestions from the reviewer and provide our response below. We hope the reviewer may also take a look at the general comments for our additional experiments, covering (1) interpretations of effective separators for each MILP class from our learning results (2) applying our method to a different MILP solver Gurobi, and (3) applying our method to a different metric (relative gap improvement under a fixed time). We hope that the reviewer may consider increasing our score accordingly if the new results adequately address the concerns.
>
> > Whether the studied part is critical enough in branch-and-bound is an important question.
>
> Understanding what separator families are useful for each MILP class is important to the mathematical programming community [1, 2] (also, see reviewer wvgK’s summary of our paper’s strengths). While a few separator families are known to be effective for specific MILP classes (e.g. Clique for Independent Set and FlowCover for problems with network flow substructures [2]), the knowledge remains limited and cannot cover the wide variety of separator families and MILP classes.
>
> Despite abundant research on cut selection, there is a surprising lack of work on the higher-level decisions such as configuring separators, or deciding whether to cut (B&C) or not (pure B&B) [3]. We hypothesize that this scarcity is because the abstract nature of the tasks poses challenges to acquiring useful heuristic information. Our work allows an automatic approach to detect effective separators for each MILP class, providing an important step toward understanding the interactions among separators and identifying when a specific separator is useful. It also allows for instance-level specification, enabling different configurations on different MILP instances. We refer the reviewer to General Response [GR1] for visualizations and interpretations of our learning results.
>
> *[1] Contardo, Claudio, Andrea Lodi, and Andrea Tramontani. "Cutting Planes from the Branch-and-Bound Tree: Challenges and Opportunities." INFORMS Journal on Computing 35.1 (2023).*
>
> *[2] Dey, Santanu S., and Marco Molinaro. "Theoretical challenges towards cutting-plane selection." Mathematical Programming 170 (2018).*
>
> *[3] Berthold, Timo, Matteo Francobaldi, and Gregor Hendel. “Learning to use local cuts." arXiv preprint arXiv:2206.11618(2022).*
>
> > First, the authors can compare the focused task, i.e., separator configuration, with other solving stages to demonstrate its significance. Or the authors can compare the resource usage to show the proposed method is lightweight enough so that it can be used jointly with other methods.
>
> **The Immediate and Multi-step Effect of Separator Configuration in the B&C Process:**
>
> 1. immediate: some separators take a long time to run, but generate mostly low-quality cuts that are ultimately never selected by the downstream cut selector. Deactivating those separators leads to an immediate time improvement by reducing the time to generate the cut pool.
> 2. multi-step: improved separator configuration can tighten the dual bound faster through better-selected cuts; it may also accelerate other B&C components such as branching (e.g. strong branching requires solving many children LPs and hence may benefit from tighter dual bounds).
>
> The following table presents the total solve time and total separator execution time for our complete method L2Sep and SCIP default on several MILP classes. We report the median and standard deviation evaluated on 100 instances for each class. L2Sep significantly reduces the total separator execution time. Upon closer examination, we find L2Sep adeptly deactivates expensive yet ineffective separators while activating effective ones.
>
> ||Total Solve Time (L2Sep)|Total Separator Execution Time (L2Sep)|Total Solve Time (SCIP Default)|Total Separator Execution Time (SCIP Default)|
> |-|-|-|-|-|
> |Comb. Auc.|**0.65s (2.36s)**  |**0.02s (0.054s)**|3.01s (4.60s)   |1.39s (1.14s)|
> |Indep. Set|**3.81s (116.42s)**|**0.35s (9.94s)** |13.16s (120.89s)|7.38s (37.94)|
> |NNV|**20.75s (18.56s)**|**0.16s (0.13s)** |34.76s (25.05s) |6.58s (8.05s)|
> ||
>
> In Fig. 3 of the rebuttal pdf, we plot the primal-dual bound curves (median and standard error) of L2Sep and SCIP default on Independent Set and NN Verification. The significantly faster dual bound convergence of L2Sep demonstrates the multi-step effect of improved separator configurations. We further summarize the synergistic interaction effects between separator selection and other B&C components (branching, dual LP) in the next table. Notably, even though our method does not modify branching, the branching solve time is reduced.
>
> ||Strong Branching Time|Pseudocost Branching Time|Dual LP Time|
> |-|-|-|-|
> |Comb. Auc. (L2Sep)|0.31s (1.15s)|0.35s (1.46s)|0.08s (0.58s)|
> |Comb. Auc. (SCIP Default)|0.41s (1.79s)|0.45s (2.17s)|0.18s (0.83s)|
> |Indep. Set (L2Sep)|2.82s (21.44s)|3.01s (22.29s)|0.3s (73.95s)|
> |Indep. Set (SCIP Default)|3.92s (19.68s)|4.6s (21.67s)|1.14s (55.51s)|
> |NNV (L2Sep)|6.56s (4.06s)|8.18s (4.81s)|3.67s (6.52s)|
> |NNV (SCIP Default)|8.31s (5.55s)|9.69s (6.13s)|5.07s (5.81s)|
> ||
>
> **Resource usage of learned separator configuration v.s. other B&C parts (cut selection, branching):** At inference time, we require only two model calls (two separator configurations) during each MILP solve. In contrast, learning to branch or cut selection requires model calls at a higher frequency, such as at each node of the B&C tree (branching) or at each separation round (cut selection). Moreover, each model call for cut selection requires evaluating each cut in the cutpool ($\approx 10^2-10^3$), whereas each of our model calls is more efficient due to the reduced number of separator configurations to evaluate ($\approx 20-30$, enabled by our restricted space). As such, our method is lightweight enough to be integrated during the solve process.

---

> > ### Comment · Reviewer_JiCg · 2023-08-15
> > **Thanks for the response and sory for the late reply. Increasing my score from 5 to 6.**
> >
> > I appreciate the authors' efforts in responding to my concerns and I'm sorry for the late reply. I have read the authors' response and other reviewers' comments. I can understand the importance of the task better now, which has addressed my main concern. I have raised my score from 5 to 6.
> >
> > Still, though the authors provide a comparison of efficiency between the proposed method and other B&C parts (e.g., cut selection, branching), I think the paper will be stronger if direct comparisons of end-to-end performances can be conduct.

---

> > > ### Author Response · Authors · 2023-08-16
> > > **Thank you! Please see authors’ further response.**
> > >
> > > We really appreciate that the reviewer takes a careful look at our rebuttal and and other reviewers’ comments. Thank you for taking the time to provide valuable feedback and revise the assessment!
> > >
> > > We provide our further responses regarding “the direct comparison of end-to-end performances” below. If our answers do not align with the reviewer intention, we would appreciate that the reviewer explain their suggestion in more details. The bottom line is that subtle differences in the emerging related works result in challenges in directly comparing without adapting and extending implementations (it is not as easy as re-running the authors’ code), not to mention that some implementations are not available. We thus offer comparisons based on reported improvement metrics for overlapping benchmarks.
> > >
> > > **What we are able to say about performance comparison:**
> > >
> > > - **Cutting:** In Sec. 4.3 of our main paper, we contextualize our performance for separator configuration with prior cut selection works on comparable datasets, where L2Sep achieves a 37.5% time improvement on NNV and 12.9% on MIPLIB (a larger subset), whereas prior cut selection works report 11.67% on NNV and 3% & 1% on MIPLIB (two smaller subsets).
> > > - **Branching:** A recent work [4] compares a few learned branching rules with SCIP default on the Ecole datasets, and report the best time improvements of 36.4% (Comb. Auc.), 34.1% (Indep. Set), and 57.4% (Fac. Loc.) from SCIP default (see Table 2 in their Appendix). In contrast, our method L2Sep achieves 66.2% (Comb. Auc.), 72.4% (Indep. Set) and 29.4% (Fac. Loc.) on larger and more heterogeneous Ecole instances.
> > >
> > > **Challenges in direct comparisons:** It is challenging to directly compare with prior learning works due to differences in experimental setup. For example, [1,2] focuses on the comparing different cutting plane selection strategies (including learning) on a synthetic environment that consider pure cutting plane iterations (without branch-and-bound) on Tang et al. instances. That is, they do not consider improvement relative to a full B&C solver (e.g., SCIP). Another recent work [3] is restricted to solving only the root node of B&C. In addition, prior works also consider different performance metrics (e.g. reversed Integrality-gap-closed integral), since they do not fully solve the B&C. Our work compares with the full B&C solver (SCIP and Gurobi), solves the full B&C tree, and considers the performance metrics of relative time or optimality gap improvement. The reviewer comment does highlight an emerging research gap, however, to unify and rigorously relate the emerging works in this area!
> > >
> > > We hope our response align the reviewer’s intended suggestion. We would like to thank again for the reviewer’s suggestions during the rebuttal period!
> > >
> > >
> > > *[1] Tang, Yunhao, Shipra Agrawal, and Yuri Faenza. "Reinforcement learning for integer programming: Learning to cut." International conference on machine learning. PMLR, 2020.*
> > >
> > > *[2] Paulus, Max B., et al. "Learning to cut by looking ahead: Cutting plane selection via imitation learning." International conference on machine learning. PMLR, 2022.*
> > >
> > > *[3] Wang, Zhihai, et al. ‘Learning Cut Selection for Mixed-Integer Linear Programming via Hierarchical Sequence Model’. The Eleventh International Conference on Learning Representations, 2023.*
> > >
> > > *[4] Scavuzzo, Lara, et al. "Learning to branch with tree mdps." Advances in Neural Information Processing Systems 35 (2022): 18514-18526.*

---

> > > > ### Comment · Reviewer_JiCg · 2023-08-18
> > > > **Thanks for the further response. I would like to keep my score, 6 weak accept.**
> > > >
> > > > Thanks for your further response. The performance comparison demonstrates the advances of the proposed method to some extend, and I understand the challenges in more direct comparisons. Since the idea of using learning method to replace parts of solvers is not that novel, and the advantage of the proposed method to previous ones is not sufficiently demonstrated, I would like to keep my score, 6 weak accept.

---

### Official Review · Reviewer_LDTP · 2023-07-08

**Soundness:** 3 good
**Presentation:** 1 poor
**Contribution:** 2 fair
**Rating:** 5
**Confidence:** 3

**Summary:**

This paper uses machine learning to decide which families of cutting planes should be applied in each of a finite number of rounds when a discrete mathematical optimization problem is solved by an MILP solver. The authors conduct experiments on the SCIP solver, in which they show that their selection of separators (generators of different families of cutting planes) is able to solve problems faster than the default configurations of SCIP.

**Strengths:**

Within the mathematical optimization community, it is known that many people have looked at this problem but few obtained good results, which shows in the limited number of direct references shown by the authors.

The results are certainly motivating, and the authors managed to analyze their work across a representative number of datasets.

*****

Following the rebuttal, I still have concerns about the significance of the work, which are similar to the ones presented in reviewer wvgK's review. For that reason, I cannot be more enthusiastic than a borderline accept for this paper.

**Weaknesses:**

I tend to consider the selection of separators a special case of what was done in prior work, in which the cuts themselves were selected. By using a setting in which their work is not comparable to prior ML studies on this topic, I am left wondering if what is proposed in this paper indeed leads to a better approach than the ones already known.

The paper explains very little about MILP, to the point that it fells like the application is an afterthought. While the appendix generously makes up for that, a reader unfamiliar with MILP might read the entire paper and not understand much about the application considered. In fact, for a span of approximately 3 pages (Line 56 in Page 2 to Line 175 in Page 5) there is very little that is specific to the application to MILP solvers. In great part that is because of using a language that is very different. I would not say that this is an issue of using ML instead of MILP terminology because I also got lost with the abstractions used, such as when the authors say "single configuration update" to mean that the same separators would be used in every round. This abstractness of the language left me with many questions about what exactly was done (see Questions). For someone with a greater interest in MILP as myself, I find it difficult to translate what the authors did to the application. As a consequence, I would have a hard time trying to reproduce their work if I wanted to.

**Questions:**

1) Can you compare your work with prior cut selection approaches?

2) In plain terms, what exactly is the subset of configurations A and how is it pre-selected?

3) What is the relevance of Propositions 1 and 2 in the context of MILP?

4) How exactly are the separator nodes S used in the representation of the instances?

5) How exactly do you consider instances in which neither SCIP nor skip with your separator selection were able to solve?

Figure 1: The letter b is used both as the RHS of the set of constraints as well as the RHS of the next cut generated; also using A and a with different meanings (constraint LHS and cut LHS) does not seem advisable.

Equation 2: Why is the term on the left repeated at the end of the equation? Is this a typo?

**Limitations:**

Although the authors do not explicitly acknowledge this, their focus on runtime for solving problems to optimality means that their approach is of little help precisely for the case in which it would be needed the most: problems in those benchmarks for which the provable optimal solution is not known. It is also unclear what happens in the case of the problems that do not finishing running on time: my guess is that timeout is counted equally as bad, although it would make more sense to look for the best solution found or the remaining optimality gap.

---

> ### Author Rebuttal · Authors · 2023-08-09
>
> We thank the reviewer for valuable input on paper presentation and other insightful questions. We provide our responses next, and present our new experiments in the general comments, covering (1) analysis of effective separators for each MILP class from our learning results, and applying our method to (2) a different MILP solver Gurobi, and (3) a different metric (relative gap improvement). If appropriate, we encourage the reviewer to increase our review score.
> > The paper explains very little about MILP, to the point that it feels like the application is an afterthought
>
> As the camera ready permits one more page, we plan to present more details on MILPs (which are currently in the Appendix) and provide additional clarification on other terminologies.
> > Q1. Can you compare your work with prior cut selection approaches?
>
> **Learning Method differences:** prior cut selection learning methods do not consider reducing the dimensionality of the action space. However, our separator configuration space with a size $2^M$ is challenging to learn (see ablation “No Restr” in Table 2). This motivates our proposed data-driven configuration space restriction algorithm.
>
> **Result Comparison:** in Main Paper Sec. 4.3, we contextualize our time improvement on comparable datasets (NNV and MIPLIB); our method achieves larger time improvements than prior cut selection works.
>
> **Task differences:** Separator configuration is applied for cut generation, which happens earlier than cut selection (See Main Paper Fig. 1 caption). At each separation round, activated separators first generate cuts to a cutpool; then cut selection algorithms choose cuts from the cutpool. The two tasks are mostly orthogonal and can be combined in future work.
> > Q2. In plain terms, what exactly is the subset of configurations A and how is it pre-selected?
>
> Each element of the subset A is a combination of separators (e.g., Gomory, Clique) to activate; we call this a configuration and it can be thought of as a binary vector of length the number of separators (17 for SCIP). We select A from the full configuration space by choosing configurations that are effective but without being redundant. To do so, we use a small training set for each MILP class (100 instances) and semi-randomly generating and evaluating around 2000 configurations to select the target subset A of size around 15-30. The specific strategy leverages submodularity, i.e., diminishing marginal returns on performance with more configurations, which justifies the use of a greedy strategy to select the configuration subset A.
> > Q3. What is the relevance of Propositions 1 and 2 in the context of MILP?
>
> Our learning method first constructs a subset A of high quality separator configurations for the MILP. We then learn a network $\tilde{f}_A$ to select a configuration among A.
>
> Prop. 1 characterizes the test performance of $\tilde{f}_A$ as a function of the subset A, which sheds light on how to construct a good A: an ideal subset A allows $\tilde{f}_A$ to have (1) high training performance, obtained when **some** configuration in A achieves good performance for *any* MILP instance in a training set, and (2) low generalization gap, achieved when **each** configuration in A has good performance *across* MILP instances in a test set. In practice, we approximate the generalization gap using the training set  (see empirical justification in Appendix Fig. 5).
>
> Prop. 2 formalizes the diminishing marginal returns (submodularity) of $\tilde{f}_A$’s training performance with respect to A, which enables a greedy algorithm to iteratively construct A. Based on Prop. 1, we further augment the greedy algorithm with a filtering criterion to improve the generation gap by eliminating ineffective configurations from A.
>
> > Q4. How exactly are the separator nodes S used in the representation of the instances?
>
> Our network takes a separator configuration-MILP instance pair as input and predicts the time improvement of applying the configuration to the instance.
>
> We represent each configuration by M separator nodes; each node has M+1 dimensional input features, representing whether the separator is activated (the first dimension), and which separator it is (one-hot M-dimensional vector). Our GNN input graph connects each separator node with all variable and constraint nodes; the input features of latter nodes contain the MILP instance information (see Appendix A5.2).
> > Q5. How exactly do you consider instances in which neither SCIP nor SCIP with your separator selection were able to solve?
> >
> >
> > In the case of the problems that do not finishing running on time, it would make more sense to look for the best solution found or the remaining optimality gap.
>
> For larger MILP classes, we exclude instances that cannot be solved by SCIP default to a specific optimality gap within a predefined time limit, ensuring most instances are retained (optimality at 120s for NNV, and a gap of 10% at 300s for MIPLIB and Load Balancing, see Appendix A.6.5); 26% of NNV instances are excluded while all instances from Load Balancing are retained. Our experiments then consider the time improvement of all instances to reach the respective gaps. In general, we can increase the gap threshold to allow learning on harder MILP classes.
>
> We conduct a new experiment where we change the objective to the relative gap improvement, which is amenable to instances that cannot be solved to optimality under a given time limit. For example, on Load Balancing where SCIP default has a average gap of 32% within 16s, L2Sep is able to reduce the average gap to 21% (34% improvement). We refer the reviewer to General Response [GR3] for details.
> > Eq. 2: Why is the term on the left repeated at the end of the equation?
>
> The test performance $\Delta(\tilde{f}_A)$ on an unknown test set can be decomposed into (1) the training set performance $\hat{\Delta}(\tilde{f}_A)$, and (2) the generalization gap $\hat{\Delta}(\tilde{f}_A) - \Delta(\tilde{f}_A)$.

---

> > ### Comment · Reviewer_LDTP · 2023-08-13
> >
> > I appreciate the response from the authors. I did not see a comment regarding Figure 1, but I hope that this gets correct.
> >
> > In revising my score, I am counting on the word of authors that a final version would be more informative about the application (MILP), and I also second reviewer wvgK's recommendation that the paper title should be changed to better reflect what the paper does.

---

> > > ### Author Response · Authors · 2023-08-13
> > > **Authors' Further Response to Reviewer LDTP**
> > >
> > > We're thankful that the reviewer took the time to revise their assessment! We will incorporate more information about MILP and make the title change in the final version. We did not provide a comment on Fig. 1 due to the character limit of our initial response, but we will update a and b to $\nu$ and $\omega$ in the Figure (e.g. $\nu_{11}^\intercal x \leq \omega_{11}$). Thank you for your suggestions!
> > >
> > > Regarding the reviewer’s concern on the significance of our work (similar to the ones in reviewer wvgK's review), we invite the reviewer to elaborate on their concern more explicitly, so that we can use this discussion period to further address.
> > >
> > > As we are delighted that our response addressed reviewer wvgK’s concern, we thought of sending you a summary of the significance of our work. We hope this message will help the reviewer LDTP in contextualizing our work’s significance.
> > >
> > >
> > > - **Task**: we agree with reviewer TfAQ and wvgK’s response that it is important to understand “when to cut”, which is much less explored (though equally crucial) as the “how to cut” question. Separator configuration and the associated cut generation play a vital role in the B&C process; properly configuring separators can accelerate MILP solvers by 25%-70% (also, see our response to reviewer JiCg *“The Immediate and Multi-step Effect of Separator Configuration in the B&C Process”*). We are excited about our work’s potential to inspire more future studies on this “when to cut” question.
> > > - **Interpretability**: from [GR1], our learning method L2Sep automatically discovers known facts from literature regarding the efficacy of difference separators for each MILP class (where the existing literature has spent decades of efforts to initially discover these facts). L2Sep can potentially speed up the knowledge discovery process by suggesting efficient MILP class-separator family pairs for future theoretical inspection.
> > > - **Method**: different from prior cut selection works, we propose a data-driven subspace restriction algorithm followed by a learning method to configure separators. Our work integrates empirical methodology with theoretical justification, bridging gaps in the existing learning for MILP literature that has been predominantly empirical; our theory directly informs our empirical subspace restriction algorithm, whereas prior theoretical works on parameter configuration cannot (see our response to reviewer wvgK for details).
> > > - **Performance**: our learning method L2Sep for separator configuration is able to accelerate multiple MILP solvers (SCIP, Gurobi [GR2]) under different objectives (time improvement, gap improvement [GR3]) on various datasets (standard, large-scale [our paper]), demonstrating effectiveness of our method in accelerating MILP solvers.
> > >
> > > We would like to thank the reviewer again! We really appreciate your detailed feedback on the paper presentation and your other insightful suggestions.

---

### Author Rebuttal · Authors · 2023-08-09

# General Responses to All Reviewers
We thank each of the reviewers for their detailed and constructive comments. We provide the following additional interpretations and new experimental results in response to the reviewers’ suggestions.

$\ $
### [GR1] Interpretation analysis: The learned model recovers known facts from the literature regarding effectiveness of different separators

Reviewer wvgK inquired about interpretations of our learned model to understand the efficacy of different separators for different problem instances. In Rebuttal PDF Fig. 1 and 2 (see figure captions for explanations of the figures), we provide an investigation, summarize the findings below, and will include additional analysis for other MILP classes in the appendix.

**Bin Packing:** It is known that instances with few bins approximates the Knapsack problem (Clique cuts are known to be effective [1]), and that instances with many bins approximates Bipartite Matching (Flowcover cuts can be useful [2]). We analyzed the separators activated by our learned model when we gradually decrease the number of bins, and observe that the prevalence of selected Clique and Flowcover cuts increased and decreased, respectively. This is illustrated in Fig. 1, right column.

**Other MILP classes:** We provide visualizations for Independent Set and MIPLIB in Fig. 2. Clique is known as an effective separator for Independent Set [3]; L2Sep automatically recovers this fact by frequently selecting configurations that activate Clique. Meanwhile, we see that L2Sep discovers the instance heterogeneity of MIPLIB, resulting in a more dispersed distribution of selected configurations.

*[1] Boland, Natashia, et al. "Clique-based facets for the precedence constrained knapsack problem." Mathematical programming 133 (2012).*

*[2] Van Vyve, Mathieu. "Fixed-charge transportation on a path: Linear programming formulations." International Conference on Integer Programming and Combinatorial Optimization. (2011).*

*[3] Dey, Santanu S., and Marco Molinaro. "Theoretical challenges towards cutting-plane selection." Mathematical Programming 170 (2018).*

$\ $
### [GR2] Learning-to-separate effectively accelerates state-of-the-art MILP solver Gurobi

As inquired by Reviewer wvgK, we replicated our method L2Sep with Gurobi (containing a larger set of 21 separators). As Gurobi is closed source, we cannot change configurations after the solving process starts$^1$, so we only consider one stage of separator configuration.

To our delight, L2Sep achieves significant relative time improvements over the Gurobi default, with gains ranging from 12% to 56%. This result confirms the efficacy of L2Sep as an automatic instance-aware separator configuration method. Similar to our results for SCIP, we observe that (1) our two heuristic sub-components (See Section 4.1) achieve impressive speedup from Gurobi default, indicating the high quality of our restricted configuration subspace, and (2) our complete method L2Sep improves the performance further, highlighting the benefit of learning instance-aware configurations. Our results (median and standard deviation) are as follows:
||Method|Max. Cut|Pack.|Comb. Auc.|Fac. Loc.|
|-|-|-|-|-|-|
||Default Time (s)|0.087 (0.051)|4.048 (3.216)|1.687 (3.596)|27.872 (14.733)|
|Heuristic Baseline|Gurobi Default|0%|0%|0%|0%|
||Random|18.6% (49.0%)|15.5% (28.2%)|-10.7% (69.1%)|13.4% (46.0%)|
|Ours Heuristic Variants|Inst. Agnostic Configuration|35.1% (35.8%)|22.9% (39.4%)|3.1% (65.3%)|40.6%  (48.1%)|
||Random within Rest. Subspace|37.3% (48.0%)|24.3% (32.2%)|5.1% (84.2%)|40.2% (46.8%)|
|Ours Learned|L2Sep|**45.4% (38.4%)**|**30.6% (29.6%)**|**12.6% (63.5%)**|**56.7% (35.7%)**|
||

*$^1$ Gurobi official documentation states “Parameters control the operation of the Gurobi solvers. They must be modified before the optimization begins.”*


$\ $
### [GR3] Learning-to-separate is effective under alternative objective (relative gap improvement)

Inspired by reviewer LDTP’s comments, we analyzed an alternative objective of the relative gap improvement under a fixed time limit. Let $g_0(x)$ and $g_\pi(x)$ be the primal-dual gaps of instance $x$ using the SCIP default and another configuration strategy $\pi(x)$ under a fixed time limit $T$. We define the relative gap improvement as $\delta_g(\pi(x), x) := (g_0(x) - g_\pi(x)) / (\max(g_0(x), g_\pi(x)) + \epsilon)$. We choose the denominator to avoid division by zero when the instance is solved to optimality.

In the table below, we find that L2Sep achieves a 15%-68% relative gap *improvement* over SCIP default. Specifically, the table presents the relative gap improvement (mean and standard deviation) of each method over SCIP default, along with the fixed time limit for various MILP classes (mostly around 50% of medium SCIP default solve time), and the absolute gap of SCIP default at the time limit. In Rebuttal PDF Fig. 3 (right two columns), we further plot histograms of the gap distribution on the entire dataset for L2Sep and SCIP default, where we observe L2Sep effectively shifts the *entire gap distribution* to a lower range. These results demonstrate the effectiveness of our method across different objectives, and its ability to improve primal-dual gaps for instances that cannot be solved to optimality within a given time limit.
||Method|Pack.|Comb. Auc.|Indep. Set|NNV|Load Balancing|
|-|-|-|-|-|-|-|
||Time Limit (s)|4.4|1.4|8.2|16|16|
||Default Gap|9.1e-4 (9.3e-4)|0.060 (0.098)|0.057 (0.059)|0.50 (0.80)|0.32 (0.13)|
|Heuristic Baseline|SCIP Default|0%|0%|0%|0%|0%|
||Random|-37.1% (41.7%)|-27.3% (69.1%)|-23.2% (44.1%)|-40.3% (72.8%)|-48.0% (35.8%)|
|Ours Heuristic Variants|Inst. Agnostic Configuration|11.9% (38.4%)|52.4% (45.3%)|23.5% (34.5%)|33.6% (72.1%)|14.0% (18.9%)|
||Random within Rest. Subspace|10.1% (42.5%)|54.1% (45.1%)|21.6% (33.7%)|24.8% (75.9%)|9.5% (17.6%)|
|Ours Learned|L2Sep|**15.4% (40.0%)**|**68.8% (38.2%)**|**29.6% (34.7%)**|**36.0% (68.2%)**|**34.2% (27.5%)**|
||

---

### Decision · Program_Chairs · 2023-09-21

**Decision:**

Accept (poster)

**Comment:**

The paper has received mixed reviews that improved partially in the course of the rebuttal process. The good rebuttal has been noted. On the upside reviewers note impressive results and nice theoretical results as well as good writing. On the more pessimistic side, limited novelty and inclarities were noted.
The authors are urged to follow wvgK's recommendation for changing the title and other suggestions that have come up during the rebuttal process.
In summary, this paper has met the bar for acceptance for NeurIPS.